# A distant global control region is essential for normal expression of anterior *HOXA* genes during mouse and human craniofacial development

Andrea Wilderman [1], Eva D'haene [2], Machteld Baetens[2], Tara N. Yankee[1], Emma Wentworth Winchester [1,3], Nicole Glidden[4], Ellen Roets[5], Jo Van Dorpe [6], Sandra Janssens[2], Danny E. Miller[7,8,9,10], Miranda Galey [7,9], Kari M. Brown[11,12], Rolf W. Stottmann [13,14,15], Sarah Vergult [2], K. Nicole Weaver[11,16], Samantha A. Brugmann [11,12], Timothy C. Cox [17,18] & Justin Cotney [4,19] ✉

Craniofacial abnormalities account for approximately one third of birth defects. The regulatory programs that build the face require precisely controlled spatiotemporal gene expression, achieved through tissue-specific enhancers. Clusters of coactivated enhancers and their target genes, known as superenhancers, are important in determining cell identity but have been largely unexplored in development. In this study we identified superenhancer regions unique to human embryonic craniofacial tissue. To demonstrate the importance of such regions in craniofacial development and disease, we focused on an ~600 kb noncoding region located between *NPVF* and *NFE2L3*. We identified long range interactions with this region in both human and mouse embryonic craniofacial tissue with the anterior portion of the *HOXA* gene cluster. Mice lacking this superenhancer exhibit perinatal lethality, and present with highly penetrant skull defects and orofacial clefts phenocopying *Hoxa2-/-* mice. Moreover, we identified two cases of de novo copy number changes of the superenhancer in humans both with severe craniofacial abnormalities. This evidence suggests we have identified a critical noncoding locus control region that specifically regulates anterior *HOXA* genes and copy number changes are pathogenic in human patients.

Proper control of gene expression during development and in adult tissues is achieved in part through regulatory sequences typically referred to as enhancers. Enhancers are collections of transcription factor binding sites that have been shown to control gene expression in a temporal and tissue-specific manner[1,2]. Some genes are regulated by a single enhancer in a particular tissue or context[3], but most genes are regulated by multiple enhancers with each contributing to a portion of the overall target gene expression[4]. Coactivated clusters of individual enhancer elements and co-regulation of multiple nearby genes has been seen as a strong biomarker for cell and tissue-type specification genes. These activated regulatory landscapes, referred to as superenhancers, are frequently associated with genes for cell-type specific transcription factors, giving them an important role in determining cell identity[5,6]. Due to their size, sequence composition, and

largely unknown contributions to specific gene regulation, the impact of noncoding mutations and copy number variation in superenhancers presents a complicated area of study. Thus far copy number changes within superenhancer regions have been associated with tumorigenesis[5], and disease-associated SNPs are enriched in super-enhancers active in cell types relevant to the disease[7]. While super-enhancer regions are also potential regulators of early developmental processes, their role in developmental defects has yet to be clearly established. Isolated clinical reports have indicated that insertion of a superenhancer into a new context can result in craniofacial abnormalities[8] but how frequently this occurs is unknown.

Superenhancers have been identified via active chromatin marks combined with a high degree of occupancy by master regulators including the Mediator complex[6,9]. Whether superenhancers con-stitute a unique paradigm for gene regulation has been a question since their definition[10–12]. Tissue-specific superenhancers are clusters of individual enhancer modules appearing to be specifically co-activated and potentially operating as a unit in a specific tissue. How-ever, this does not preclude the activity of individual enhancer ele-ments in more than one developmental time or tissue. The behavior of individual enhancer elements varies within superenhancers, with some studies suggesting they cooperate in an additive, redundant, or even synergistic manner[4,11,13–15]. Individual enhancer elements can also dis-play redundant functions within the larger regulatory context. For instance, deletion of individual deeply conserved enhancers within a superenhancer surrounding *Arx*[9] resulted in minimal phenotype, and it was only after two or more modules were perturbed was a strong effect on gene expression observed[16]. Other studies have characterized superenhancers in which a single potent enhancer element drives the majority of the effect[15,17]. Interestingly the potency of a super-enhancer or the individual enhancer elements within it cannot be definitively predicted by degree of conservation[4,16,18,19]. Therefore, a super-enhancer may best be studied as a complete unit.

The major challenges in the study of superenhancers are twofold. First, their specificity requires that superenhancers be identified in the relevant tissue at the developmental stage of interest. Second, per-turbation of a superenhancer and downstream consequences may only be apparent during developmental or specific conditions that are difficult to create experimentally[20,21]. Despite functional annotations of active chromatin states being available for numerous human tissues, early developmental stages are underrepresented and therefore many superenhancers have yet to be identified.

We previously used ChIP-seq for six histone modifications cou-pled with imputation of additional epigenomic characteristics to pro-vide epigenomic annotations during critical stages of human craniofacial development[22]. This approach revealed that variants associated with common nonsyndromic craniofacial abnormalities such as nonsyndromic cleft lip with or without cleft palate (NSCL/P) are enriched in enhancers active in early developmental stages. In con-trast, we and others have subsequently shown with this data that var-iants associated with normal facial variation are enriched in enhancers active in later developmental stages[22–24]. The comprehensive nature of the data obtained from our previous investigation makes it ideally suited for the identification of craniofacial-specific superenhancers and investigation of their role in nonsyndromic craniofacial malformations.

Here, we report novel human craniofacial-specific superenhancer regions in developing craniofacial tissue spanning organogenesis and describe their general characteristics. Additionally, we identified craniofacial-specific superenhancers that do not harbor known genes and tested the function of one such region. Our examination identified a novel superenhancer region that interacts with the *HOXA* locus in human and mouse embryonic craniofacial tissues. We demonstrate that deletion of this novel superenhancer in mice decreases anterior

*Hoxa* gene expression in pharyngeal arch tissue and recapitulates the distinct craniofacial phenotypes reported in *Hoxa2* null mice. We include discussion of patients with copy number variations over-lapping this region and the potential for pathogenicity of noncoding mutations in this region in humans.

## Results

### Identification of novel craniofacial superenhancers from epigenomic atlas

We hypothesized that groups of enhancers co-activated during cra-niofacial development might be enriched for novel master regulator genes as well as regions of the genome likely to be linked to cranio-facial abnormalities. To address this hypothesis, we first sought to identify superenhancer regions in a systematic fashion across cranio-facial development. Using 75,928 previously identified craniofacial enhancer segments and H3K27ac ChIP-seq signals from 17 samples of human craniofacial tissue across five embryonic and one fetal stage encompassing the major events in craniofacial development[22] we called superenhancer regions genome-wide using the ROSE algorithm (see Methods[6]). We identified an average of 1861 superenhancers per sample with a total of 4,339 distinct superenhancer regions across the developmental trajectory (Supplemental Table 1). The superenhancers calls were generally found to separate into two major groups: those that are active early during Carnegie Stages (CS) 13 – 15 and those that are active later in development CS17-fetal (Figure S1). The super-enhancers identified in craniofacial tissues ranged in size from a few thousand base pairs to more than 550 kilobases (kb), with median and mean sizes of 52.6 kb and 58.1 kb respectively. These distributions were very similar to superenhancer calls available for 20 other human tissues based on H3K27ac as calculated by dbSuper[9] (Figure S2A).

Superenhancers based on H3K27ac signals typically encompass their tissue-specific targets or reside in the introns of their target genes[5,6]. Consistent with this, approximately 90% of individual cra-niofacial superenhancers overlap at least one gene (average 2.3+/−0.22 protein-coding genes per embryonic sample, Mean +/− SD). We found similar results for superenhancers identified in tissues profiled by the Roadmap Epigenomics and ENCODE projects retrieved from dbSuper[9] (Figure S2B; Supplemental Table 2). These embryonic craniofacial superenhancers often overlapped bivalent promoters of DNA binding factors, including many homeobox transcription factors[22]. Genes encompassed by these craniofacial superenhancers were also sys-tematically enriched for craniofacial disease related ontologies including craniofacial abnormalities, abnormalities of the midface, abnormalities of eyes (e.g., exophthalmos), and terms for facial char-acteristics (e.g., frontal bossing, pointed chin, round face) (Supple-mental Table 3)

### Characterization of craniofacial specific superenhancers

Having demonstrated that our craniofacial superenhancer calls were consistent in both size and scale to other human tissues and were enriched for craniofacial relevant biology, we next investigated whe-ther any superenhancer calls represented clusters of individual enhancer segments with a pattern of co-activation specific to cranio-facial development. To achieve this we determined intersections of the 4339 distinct craniofacial superenhancer regions with all super-enhancer calls from dbSuper as well as the human embryonic heart[25], representing the only other publicly available set of superenhancers from a human embryonic tissue. The majority of superenhancer calls, 3459, were shared with at least one other human tissue in dbSuper, 349 superenhancers were identified more generally in embryonic development, and 531 superenhancer regions were only called in cra-niofacial tissue (Fig. 1a).

The craniofacial-specific superenhancers (CSSEs) were smaller than all craniofacial superenhancers (median 37.4 kb vs. 52.6 kb) and

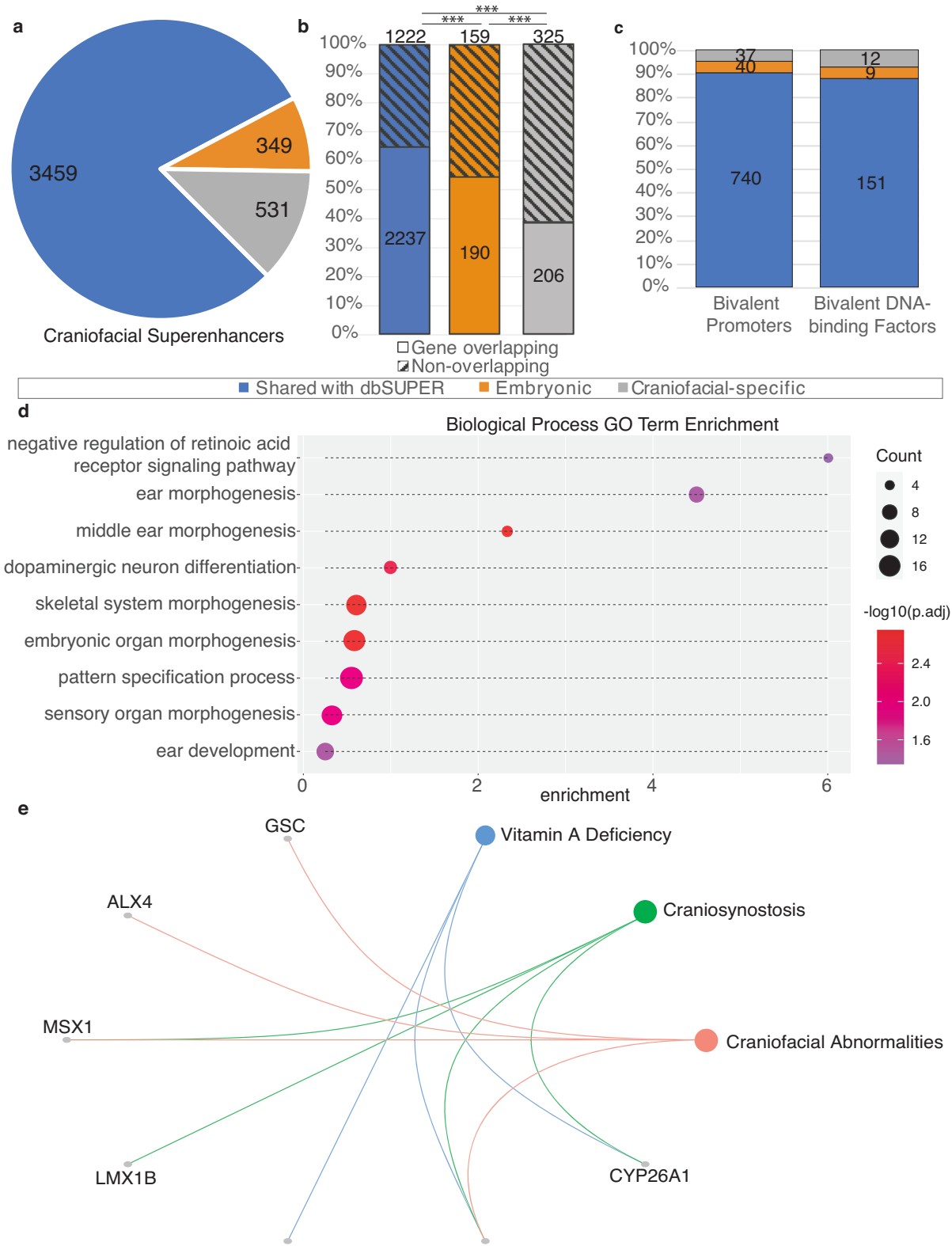

contained fewer genes (Supplemental Table 4). Significantly fewer CSSEs encompassed the transcription start site (TSS) of a gene than those identified in dbSuper and shared with the embryonic heart (Fig. 1b). Many of the genes overlapped by superenhancers, including CSSEs, were previously determined to have bivalent promoters in human embryonic craniofacial tissue and included many DNA binding factors (Fig. 1c). We have previously shown such bivalent genes are enriched for known craniofacial disease genes[22]. When we examined the full set of genes encompassed by CSSEs, we found significant enrichment for functions related generally to development, including embryonic organ development and skeletal system morphogenesis. We also identified enrichment of genes related to sensory organ

**Fig. 1 | Characteristics of human embryonic craniofacial superenhancers relate to specialized developmental functions. a** Sharing of superenhancer regions (see Methods) with tissues and cell types within dbSUPER (blue), or only with human embryonic heart (termed Embryonic enhancers, orange). Those unique to the human embryonic craniofacial tissue are shown in gray. **b** Percentages of shared or unique superenhancer regions which encompass a TSS (Gene-overlapping) or not. The Pearson's chi-squared test with Yates' continuity correction (two-tailed) was used to compare the proportions of gene overlapping and nonoverlapping superenhancers, \*\*\*$p < 0.001$. (dbSuper vs. Embryonic $p = 1.818 \times 10^{-4}$, Embryonic vs. Craniofacial-specific $p = 6.443 \times 10^{-6}$, dbSuper vs. Craniofacial specific

$p$-value $< 2.2 \times 10^{-16}$). **c** Summary of TSS encompassed in each category of super-enhancer regions that correlate with genes previously identified to have bivalent promoters, and a further subset of transcription factors with bivalent promoters. **d** Gene Ontology terms enriched in genes for which the TSS is encompassed by superenhancer regions unique to craniofacial tissue. Dot size is based on $-\log_{10}$ transformed Benjamini-Hochberg adjusted hypergeometric $p$-values calculated by clusterProfiler. **e** Disease Ontology relationships with genes for which the TSS encompassed by superenhancer regions unique to craniofacial tissue are among the previously determined genes with bivalent promoters.

morphogenesis including morphogenesis of the middle ear, suggesting specialized regulatory activity in these datasets (Fig. 1d). Well known craniofacial abnormality genes identified by this analysis included *MSX1* and *ALX1* suggesting CSSE loci are generally important for craniofacial development (Fig. 1e).

Having confirmed the craniofacial relevant nature of genes encompassed by CSSEs we wondered whether CSSEs that did not contain a protein coding gene might also be important for craniofacial development or disease. The non-coding CSSEs (ncCSSEs) were an average of ~200 kb from the nearest TSS, with a maximum distance of approximately 3 Mb (Supplemental Table 5). Nearly a quarter (77/325) of ncCSSEs fell within "gene deserts," regions of 500 kb or greater that do not contain known protein coding sequences (Ovcharenko et al. 2005). When we interrogated the nearest genes flanking ncCSSEs ($n = 584$ genes, $n = 325$ regions) we observed enrichment for Disease Ontology terms related to craniofacial abnormalities, clefting, and as well as types of tumors considered neurocristopathies[26] (Figure S3). However, when we examined more general Biological Process Gene Ontology terms, we found enrichment for multiple aspects of kidney development and little relationship to craniofacial development (Figure S4; Supplemental Table 5). These findings suggest that either these ncCSSEs do not play as significant a role in craniofacial development as CSSEs that contain a gene, or the genes they regulate are located more distally.

Gene deserts have been generally shown to be rich with regulatory sequences (Nobrega et al., 2003). Such swaths of non-coding sequence could harbor many important regulatory regions for craniofacial development. However, some deserts can be deleted without significant phenotypic consequence including those directly flanking very important transcriptional regulators like *Myc* (Nobrega et al., 2004[20]). While our disease ontology results might suggest ncCSSEs

play a role in craniofacial abnormalities, there is the distinct possibility that superenhancers identified by approaches residing within a desert have no bearing on embryonic development. While 77 regions are a relatively small subset of the overall superenhancers, the overall size of the regions and challenges associated with engineering large genomic deletions prevent large-scale analysis requiring prioritization of a small number of regions to further explore. To prioritize ncCSSEs for study in development, we hypothesized that ncCSSEs that consist of several validated and active individual segments were likely to have a significant role in gene regulation. We addressed this by interrogating the VISTA enhancer database that includes thousands of individual human and mouse genomic segments tested for in vivo enhancer activity[27]. Of the gene deserts harboring craniofacial-specific superenhancers, 38 had at least one enhancer segment tested for activity and found to be positive in any tissue (Supplemental Table 6). Of these, nine had at least one enhancer segment with craniofacial activity. A gene desert located on chromosome 5 (chr5:90679176-92919042, hg19) flanked by *ARRDC3* and NR2F1 had the largest number of tested segments ($n = 18$) but only two of these were positive for craniofacial activity. The gene desert with the next most tested segments was on chromosome 16 ($n = 11$, chr16:51185278-52471916) flanked by *SALL1* and *TOX3* yielded only one segment with craniofacial activity. Following these a desert located on chromosome 7 (chr7:25,268,105-26,191,859 (hg19)) flanked by *NPVF* and *MIR28A*, contained ten individual enhancer segments tested by VISTA, six of which had craniofacial activity (Table 1).

## Gene desert on chromosome 7 contains superenhancer regions unique to embryonic craniofacial tissue

Due to the high proportion of enhancer segments with confirmed craniofacial activity, we chose to focus on the gene desert located on chromosome 7. This chromosomal segment contained three regions identified by ROSE as superenhancers active in human embryonic craniofacial tissue (Fig. 2a). The superenhancer regions between chr7:25,580,400-25,880,000 (hg19) are unique to human embryonic craniofacial tissue, not having been identified as such in human embryonic heart tissue[25] or any of the 102 human tissues and cell lines analyzed by dbSuper ([9] and https://asntech.org/dbsuper/index.php) (Fig. 2b). Human and mouse enhancer segments tested by the VISTA Enhancer Browser (https://enhancer.lbl.gov/[27]; that drove reporter expression in mouse craniofacial tissue at E11.5 fell within the super-enhancers that were not called in other analyzed tissues. We tested an additional human enhancer segment, HACNS50[28] and found it to drive strong reporter expression in mouse embryonic craniofacial and limb tissue. (Fig. 2c).

To identify target genes which might be regulated by this craniofacial-specific region, we scanned for craniofacially-relevant genes located nearby. When considering genes up to 500 kb in either direction of the gene desert, we observed *NPVF, MIR28A, OSBPL3, CYCS, C7ORF31, NFE2L3, HNRNPA2B1, CBX3* and *SNX10*, none of which have been specifically associated with craniofacial development or disease (Lee et al., 2017; Braconi et al., 2010; (Li et al., 2016). When we examined expression of all these genes in primary human craniofacial tissues and all GTEX tissues, we found similar levels across most tissues

## Table 1 | VISTA elements with positive staining in craniofacial tissue within gene deserts

| Human chromosome | start | end | VISTA IDs |
|---|---|---|---|
| chr3 | 126756235 | 127291911 | mm1516 |
| chr5 | 90679176 | 92919042 | hg952, hg1153 |
| chr6 | 18469105 | 19837616 | hg1052 |
| chr7 | 25268105 | 26191859 | hg1600, mm402, mm403, mm404, mm405, mm406 |
| chr8 | 142528837 | 143293440 | mm1584 |
| chr10 | 35931206 | 37414715 | hg1567 |
| chr16 | 65160015 | 66400524 | mm40 |
| chr16 | 73093597 | 74330672 | hg1612 |
| chr17 | 68176189 | 70117160 | mm628, mm634, mm635 |

Coordinates (hg19) of gene deserts containing craniofacial-specific superenhancers and the VISTA ID numbers of putative human (hg) and mouse (mm) enhancer elements driving reporter expression in E11.5 mouse craniofacial tissue. Mouse enhancer elements in VISTA are lifted over from mm9 to hg19.

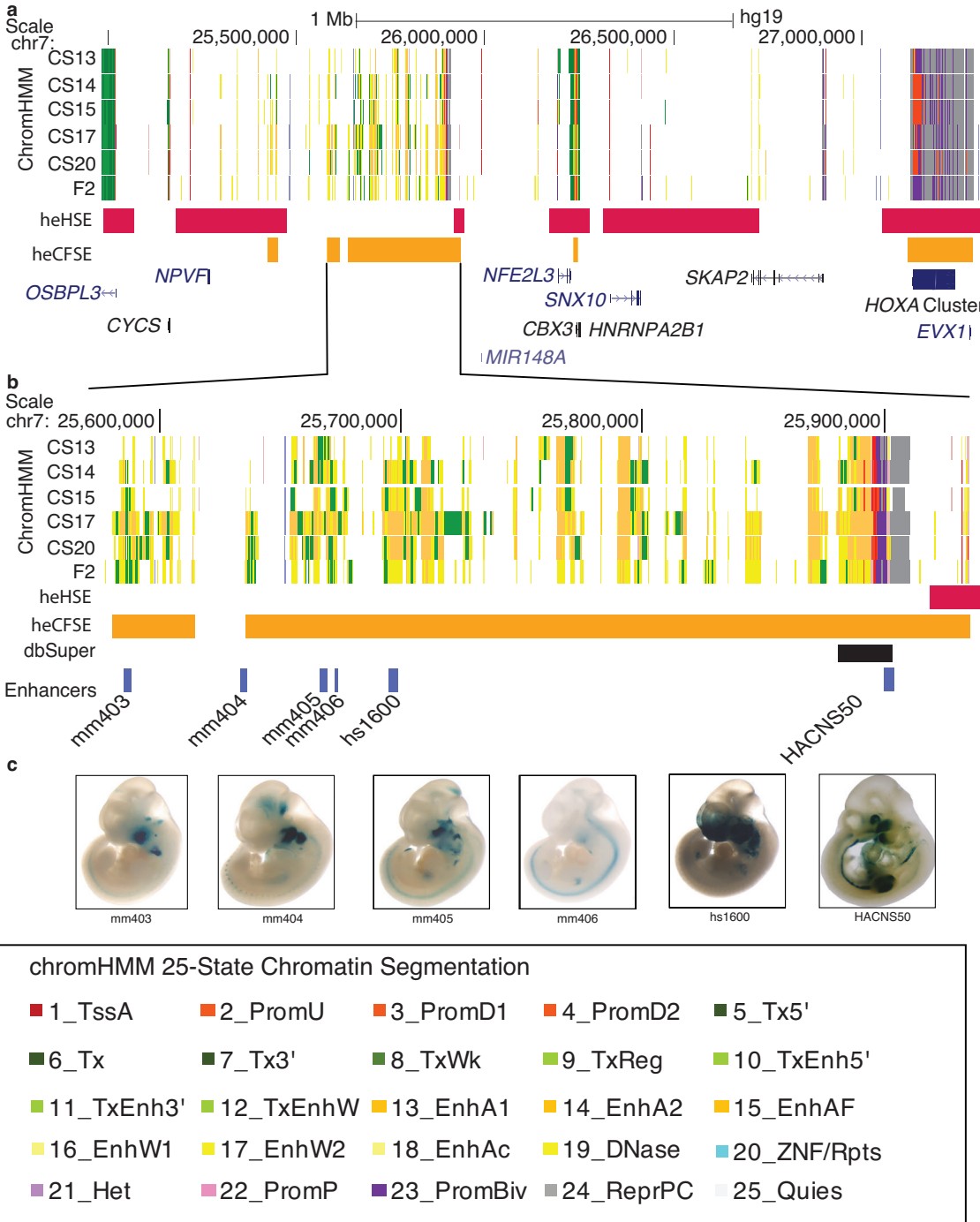

**Fig. 2 | Location and functional characterization of a putative novel craniofacial superenhancer. a** Region of human chromosome 7 containing a large 500 kb window lacking any annotated protein-coding genes with extensive enrichment of activated enhancer (yellow and orange) and transcriptionally active (green) segment annotations in human craniofacial tissue. CS (Carnegie stage). Locations of human embryonic craniofacial superenhancers (heCFSE) are represented by orange bars, human embryonic heart superenhancers (heHSE) by dark pink bars

(Figure S5[29];). Only *SNX10* had elevated specificity of expression but was most highly expressed in the adult brain. These findings raised the possibility that either one of these genes plays an unappreciated role in craniofacial development or the target of this superenhancer may lie a considerable distance away.

To test the later hypothesis, the region being considered was then expanded. This expansion revealed *SKAP2* and the *HOXA* gene

and superenhancers found in the dbSuper database by black bars. **b** Enlargement of two superenhancers with multiple validated craniofacial enhancer segments. Enhancers with mm or hs designations were identified through the Vista Enhancer Browser (**c**). In this study we tested and validated the craniofacial enhancer activity of HACNS50, located within the bivalent chromatin state at the right of the enlargement.

cluster approximately 1.5 Mb downstream. While *SKAP2* has not previously been implicated in craniofacial development, *HOXA* genes have been linked to a number of syndromes that include craniofacial abnormalities in both mouse and human[30–33]. Despite the distance between the *HOXA* cluster and the identified super enhancer, its tissue-specific relevance suggested these genes as feasible targets.

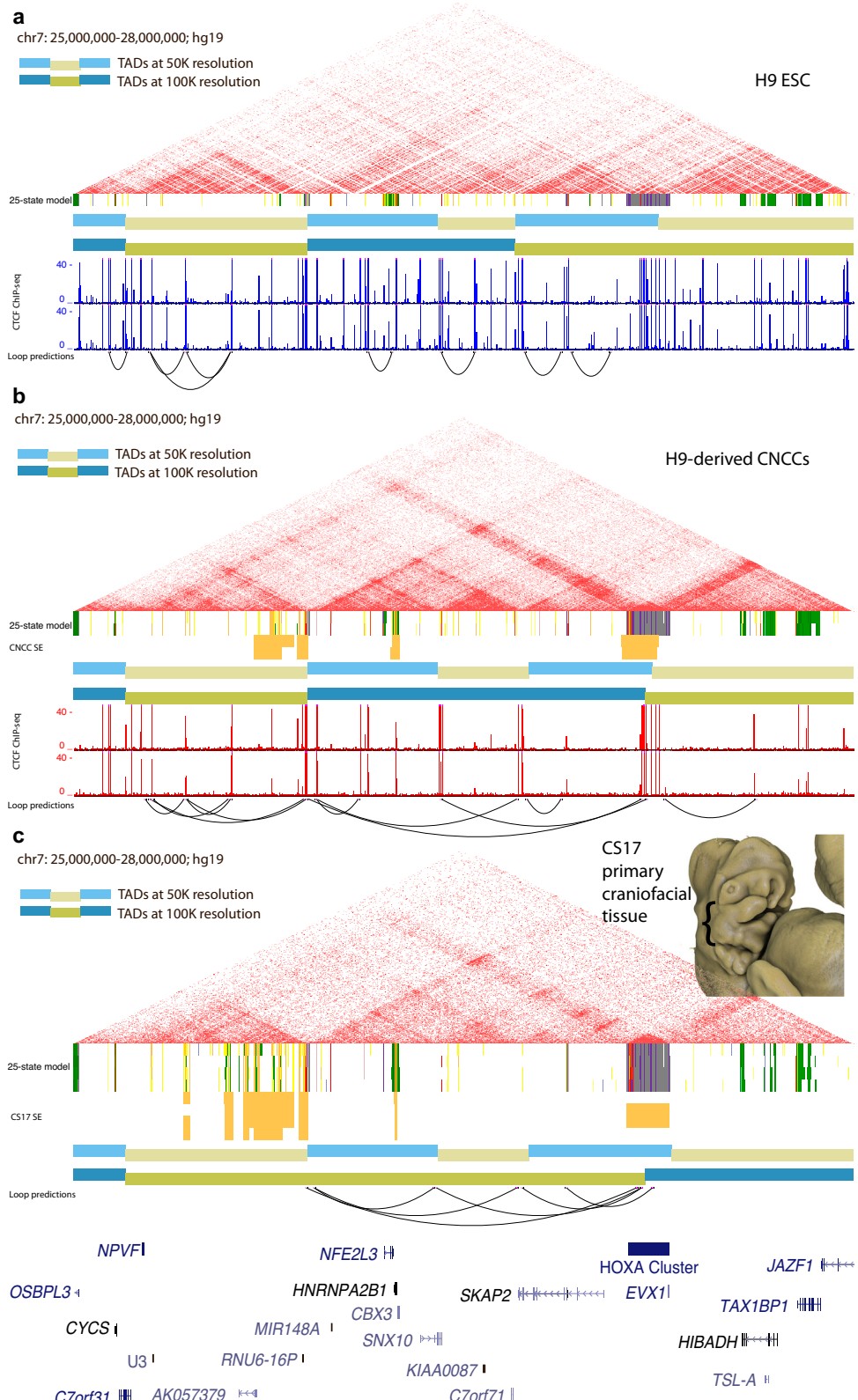

## Chromatin architecture of chr7 25,000,000–28,000,000 suggests the superenhancer makes long-distance contacts with the *HOXA* cluster

To determine if this ncCSSE region could indeed target the *HOXA* gene cluster located >1MB downstream, we first examined publicly available HiC data from a variety of cell types. In human embryonic stem cells (hESCs) we found that the entire superenhancer region formed a topologically associated domain (TAD); however, it was unclear if this region formed longer range interactions that incorporated the *HOXA* cluster (Fig. 3a). Other adult tissues and cell types showed similar trends (Figure S6), but chromatin state segmentation shows that a large portion of this region was specifically active in human craniofacial development, leaving open the possibility that the superenhancers within the TAD may drive expression only in craniofacial-specific

**Fig. 3 | Chromatin Architecture in Primary Human Embryonic Craniofacial Tissue Suggests Interaction between HOXA Gene Cluster and Gene Desert Superenhancer on Chromosome 7.** HiC of H9 human embryonic stem cells (hESCs) (**a**), cranial neural crest cells (CNCCs) derived from H9 hESCs (**b**) and CS17 primary human embryonic craniofacial tissue (**c**). TADs called at two different resolutions, 50 kb (light blue/light yellow) and 100 kb (blue/dark yellow). Super-enhancers (for CNCCs and CS17 tissue) determined by the ROSE algorithm. CTCF ChIP-seq data from (ref. 46; GSE145327). ChromHMM chromatin states from 25-state model for H9 and H9-derived CNCCs are shown below their respective HiC interaction plots and chromatin states for CS13-CS20 and F2 human craniofacial tissue are shown below the HiC interaction plot for CS17 primary craniofacial tissue. Individual enhancer segments are yellow and orange. Inset image: 3D rendered Carnegie stage 17 human embryo demonstrate representative staging of tissue used in HiC experiments The embryo was imaged using High Resolution Episcopic Microscopy (HREM): raw data courtesy of Dr Tim Mohun (Francis Crick Institute, London, UK) and provided by the Deciphering the Mechanisms of Developmental Disorders (DMDD) program (https://dmdd.org.uk/).

contexts. We therefore performed chromosome conformation capture experiments (HiC) in both primary human embryonic craniofacial tissue and a recently described culture model of differentiated cranial neural crest cells (CNCC)[34].

HiC analysis of chromosome 7 in hESC-derived CNCCs, revealed that the 3′ most boundary region of the TAD identified above formed qualitatively stronger interactions with the *HOXA* gene cluster than hESCs (Fig. 3b). However, this TAD did not coalesce into a larger domain including those genes in our analysis. Surprisingly, when we analyzed HiC data from primary CS17 (Carnegie stage 17) human embryonic craniofacial samples, a much larger TAD was discovered. This TAD stretched from the 5′-most boundary of the gene desert all the way to the midpoint of the *HOXA* gene cluster (Fig. 3c). Gene expression analysis in each of these cell types and tissues revealed that the changes in configuration relative to hESCs coincided with increased expression of *HOXA* genes (Supplemental Table 7).

## Inversion of superenhancer sub-TAD identifies TAD boundary as strong organizing center

The three-dimensional chromatin structure and associated gene expression suggested that the putative superenhancer region is part of the *HOXA* regulatory landscape[35,36]. To determine if this region is indeed important for anterior *HOXA* gene expression we set out to remove this region from the genome and examine its effects on gene expression in hESCs and CNCCs (Fig. 4a). Removal of small enhancer modules has frequently resulted in minimal effects on gene expression and mice with very mild phenotypes[18,37]. Even the disruption of large gene deserts has been reported to result in mice that are overtly normal[15,20,38]. More recently deletion of ultraconserved enhancer modules required multiple perturbations before a robust phenotype can be observed, suggesting that at minimum, pairs of enhancers may need to be disrupted to understand their function. Within the ncCSSEs of the chromosome 7 gene desert, there are as many as 215 individual enhancer modules in human craniofacial tissue. In our model system of H9 hESC-derived CNCCs, we found 72 individual enhancer modules in this locus. Creating all pairwise combinations of enhancer segment deletions in cell culture of only the CNCC enhancers in this region would require 2556 unique clones to study in this fashion. Therefore, we chose to make a single deletion of the -625 kb span between *NPVF* and *MIR148A* in H9 hESCs and differentiate to CNCCs.

When creating a deletion of such size, attention must be paid to the potential effect on regional three-dimensional chromatin structure. Disruption of TAD boundaries has been shown to result in ectopic expression of genes in the newly formed TADs[39–42], but the effects of removing an entire TAD have been less clear. We hypothesized that removal of the entire TAD containing the superenhancer might not create new TADs or alter larger chromosome architecture and could allow us to identify the specific regulatory outputs of this region. To achieve such a scenario, we aimed to identify sites for cutting by Cas9 that were outside both boundaries of the TAD.

Current models of chromosome organization indicate that CTCF is an important component for loop formation and TAD boundary establishment[43–45]. Inspection of CTCF binding in a similar model of hESC-derived neural crest cell model[46] revealed a relatively small number of occupied CTCF binding sites in this gene desert. A single strong CTCF binding event was identified at the 5′end of the TAD previously identified in our human CNCCs model. At the 3′-most end of this same TAD we observed several strong, closely spaced CTCF binding sites with the same motif orientation. Consistent with the well-documented insulating role of CTCF, these closely spaced CTCF binding sites directly coincided with the boundary between strongly active and strongly repressed chromatin signals in both CNCCs and primary craniofacial tissues (Fig. 4b).

Having identified more precise boundaries of this putative regulatory domain, we designed guide RNAs to target Cas9 to this region. On the 5′ side of the TAD we selected a sequence downstream of *NPVF* but upstream of the 5′ CTCF bound site near the TAD boundary. On the 3′ side of the TAD we selected a sequence downstream of the 3′ CTCF bound site at the TAD boundary, but upstream of another cluster of strongly bound CTCF not predicted to be part of this TAD. We then transfected plasmids expressing these guides and Cas9 protein into H9 ESCs and screened for clones that had deleted this region (Figure S7). Several clones were identified that harbored heterozygous deletions of this region. These singular perturbations did not significantly impact *HOXA* gene expression or expression of other genes in the surrounding region in CNCCs (Supplemental Table 8).

Interestingly, we did identify one clone which had lost one allele of the region and inverted the other, resulting in a hemizygous inversion illustrated in Fig. 4c (see Figure S8 for exact breakpoints). Given that previous reports of HSPCs lacking a copy of the TAD boundary had altered differentiation characteristics relative to wild type controls[47] we challenged these cells to differentiate into CNCCs. All clones we obtained grew normally in hESC culture conditions and readily differentiated to CNCCs. The number and identity of genes differentially expressed between hESC and CNCC states were very similar between the control and inversion cell line (Supplemental Table 9). When we directly compared control and inversion cell line CNCCs we found less than 100 genes were differentially expressed (Figure S9). Genes downregulated in the inversion CNCCs ($n = 52$) were enriched in pathways related to cell adhesion while those upregulated in the inversion CNCCs ($n = 44$) were enriched in pathways related to embryonic organ development and ossification (Supplemental Table 10).

While some notable putative targets of *HOXA2* were dysregulated, including *BMP4*, *MAFB*, and *FZD5*, the differentially expressed genes were not significantly enriched for Hoxa2 ChIP-seq peaks[48] (Figure S10a). Genes related to Hoxa2 signaling, determined by differential expression between E11.5 pharyngeal arches of WT and *Hoxa2*-/- mice[48] showed modest but significant enrichment in genes downregulated in the inversion CNCCs (Figure S10b). When we inspected the expression of *HOXA* cluster genes or genes surrounding the deleted region were not significantly altered in hESCs (Figure S11a). Upon differentiation in CNCCs, *HOXA3* through *HOXA9* has slightly elevated but not statistically significant changes in gene expression in the inversion line (Figure S11b). When we performed HiC in CNCCs derived from the inversion cell line we observed the strengthening of interactions between the superenhancer region and the *HOXA* gene cluster including a novel significant contact (Fig. 4d, e; Figures S12 and S13).

Despite being moved nearly 600 kb farther away from its target and the inversion of CTCF motif orientations, it was particularly surprising that contact between this TAD and the *HOXA* gene cluster was

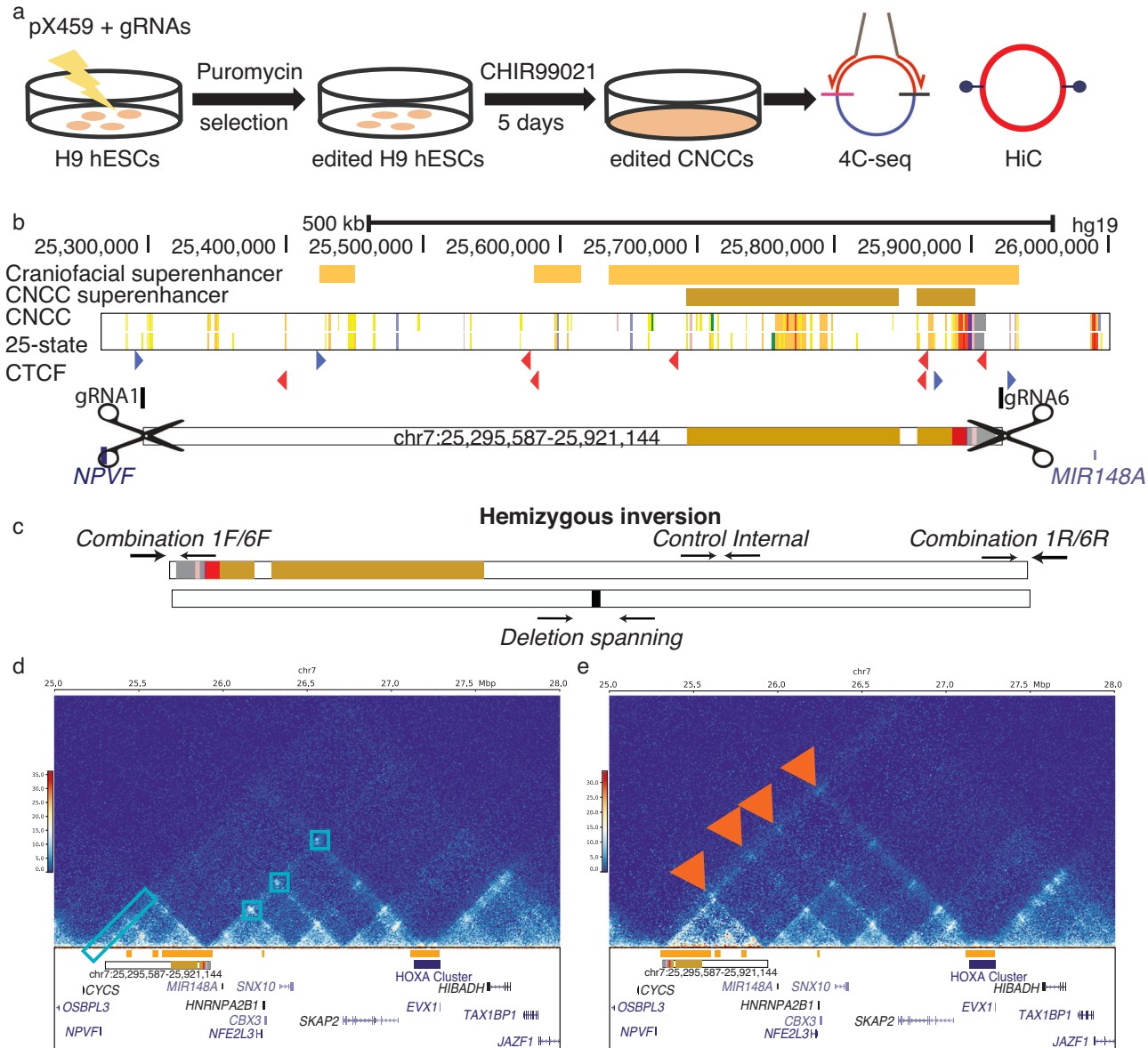

**Fig. 4 | Editing of hESCs resulted in inversion of superenhancer target.** Method of genome editing H9 cells (**a**). Location of guide RNAs gRNA1 and gRNA6 relative to the WT orientation (**b**). CTCF motifs are shown in color by orientation, blue forward and red reverse. Screening strategy for determining whether clones are heterozygous for the 1/6 deletion and determining if a clone contains an inversion of the targeted region, orientation of hemizygous inversion clone is shown (**c**). HiC interactions in H9-derived CNCCs from WT (**d**- left panel) and clone with hemizygous inversion (**d**- right panel). The HiC plot made for INV used alignment to a custom version of the hg19 genome with the specific inversion on chromosome 7 introduced. Strong contacts in WT CNCCs are marked in the left panel with light blue boxes. Novel contacts created by inversion are marked in the right panel with red arrows.

largely maintained and even enhanced. Given these results we hypothesized that *HOXA* gene expression might be maintained at normal levels in these cell lines through upregulation of the cluster that remained in *cis* with the superenhancer region. When we performed allele specific analysis of gene expression between control and inversion CNCCs, we identified allelic imbalance for variant rs2428431 within the exon of *HOXA2* transcript ENST00000612779 in inversion CNCCs (Figure S14).

**Syntenic superenhancer cluster in mouse makes 3-dimensional contacts with *Hoxa* cluster**

While these results have implications for models of chromatin architecture and loop formation, the continued expression of *HOXA* in the inversion CNCCs precluded us from making any determination of the role of this region in craniofacial related biology. We therefore asked

whether this region might have similar epigenomic properties in mice, and if it could be better studied in a system where we could generate homozygous null animals.

Syntenic regions often have preserved regulatory structure and features[43,49] and this was indeed the case for this ncCSSE. This region is part of a large syntenic block between the two species which stretches nearly 10 Mb in length with the *HOXA* gene cluster roughly at its center (Figure S15). When we compared chromatin states between human craniofacial and mouse tissues (Wentworth et al 2022) we found most similar trends in chromatin state across the gene desert in mouse craniofacial tissue (Figure S16). Chromatin state segmentation of mouse craniofacial tissue across multiple stages of development (E9.5–E15.5) demonstrated very similar patterns of chromatin activation between human and mouse across the orthologous 2 Mb region encompassing the *Hoxa* gene cluster the ncCSSEs (Figs. 5, S16).

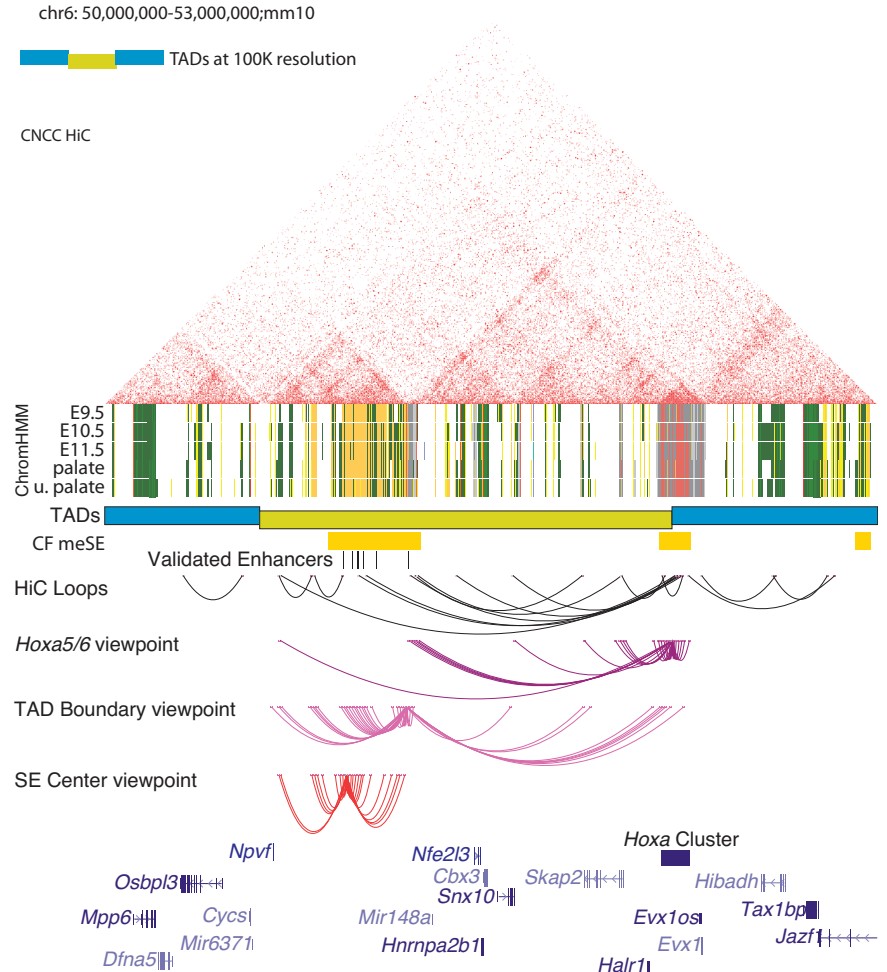

**Fig. 5 | Chromatin architecture in primary mouse embryonic craniofacial tissue.** HiC of E11.5 mouse embryonic craniofacial tissue. TADs called at 100Kb (blue/dark yellow). Mouse embryonic craniofacial superenhancers (CF meSE) determined by ROSE algorithm. Enhancer segments with validated craniofacial activity (shown in Fig. 2); human enhancer segment coordinates arrived at via liftover. ChromHMM chromatin states for the 18-state model of embryonic craniofacial tissue for E9.5- E11.5, E12.5 palate and E13.5 upper palate; individual enhancer segments are yellow and orange. HiC loops as predicted at 10Kb resolution, 4C-seq loops at 10Kb resolution (2 biological replicates per viewpoint). Viewpoint near intergenic space between Hoxa5 and Hoxa6 (purple), viewpoint at TAD boundary (magenta), and viewpoint at center of superenhancer subTAD (red).

Superehancer analysis using the same ROSE pipeline as for the human data identified an orthologous region between *Npvf* and *Nfe2l3* (Fig. 5).

Having demonstrated conservation of chromatin states during craniofacial development, we next wondered if the three-dimensional structure of the region was also conserved. Utilizing circularized chromosome conformation capture with sequencing (4C-seq[50]) we assessed the interactions of four viewpoints in this window in E11.5 mouse craniofacial tissue. For two viewpoints we identified extensive interactions within the identified window that do not cross the putative TAD boundary. When we assessed viewpoints flanking the TAD boundary, one of which contained the active enhancer HACNS50, we observed interactions within this identified region as well as significant interactions with the *Hoxa* gene cluster (Fig. 5). To confirm these interactions, we performed additional 4C-seq experiments utilizing viewpoints located directly within the *Hoxa* cluster and the promoter of the *Skap2* gene. We observed strong interactions between both viewpoints and the TAD boundary of the original window (Figure S17). Interestingly, *Hoxa* made contacts with the outer limits of this window, but contacts were not observed within the window. These findings illustrated that the region orthologous to that identified in human craniofacial tissue makes strong contacts over nearly 1.5 Mb with genes of the *Hoxa* cluster in developing mouse craniofacial tissue. Overall,

these findings indicated this region was conserved at multiple levels: primary sequence within the superenhancer region, genomic position relative to potential regulatory targets, chromatin activation in tissues across developmental stages, and long-range three-dimensional contacts.

## Deletion of the *Hoxa* global control region (GCR)
Conservation of several functional aspects pointed to this region being highly important for regulation of the *Hoxa* gene cluster and similar to a global control region (GCR) for the *Hoxd* in limb development ([36], Spitz and Duboule 2008). An x-ray induced rearrangement that placed the *Hoxd* gene cluster 700 kb further away from its GCR resulted in mice lacking an ulna (Davisson, M. T., and B. M. Cattanach 1990[36]). Therefore, the ncCSSE we identified is a potential GCR and we hypothesized disruption of this region or its ability to target *Hoxa* genes would lead to strong craniofacial phenotypes. We hereafter refer to this region as the *Hoxa* GCR. We identified guide RNAs in the mouse genome very close to the orthologous positions utilized in the human cells (Fig. 6a; see Methods). These guides were then injected with Cas9 mRNA into fertilized mouse eggs. This resulted in one heterozygous founder which was genotyped by PCR and DNA sequencing, then bred to produce the F1 generation. Five heterozygous F1 mice (3 female, 2 male) were then

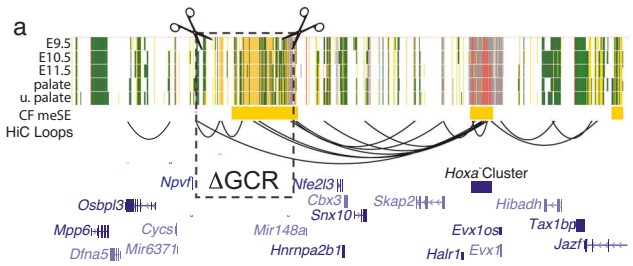

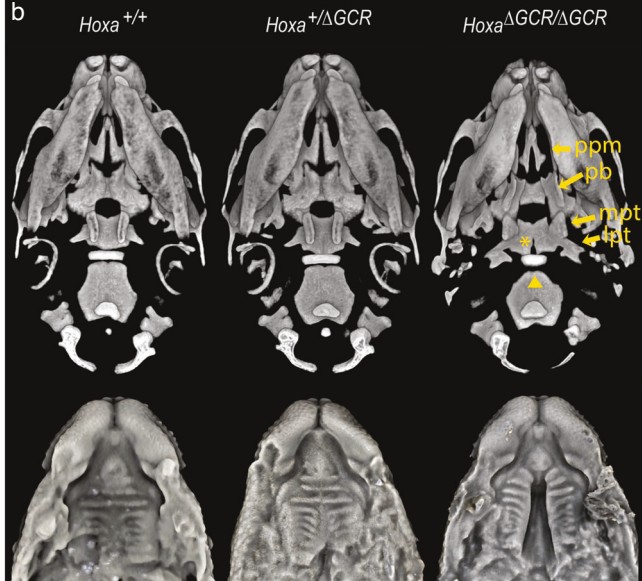

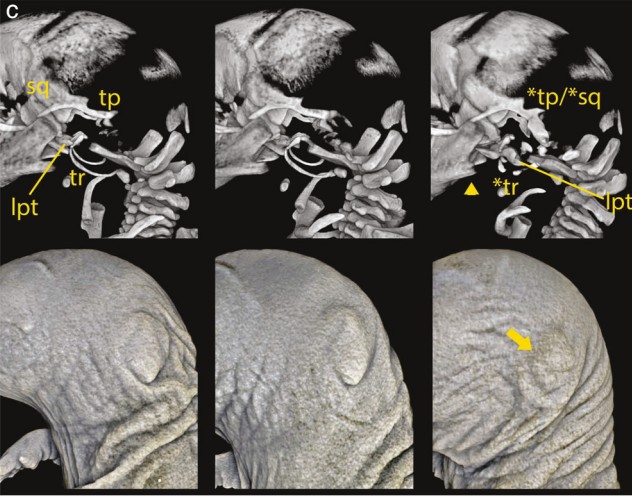

**Fig. 6 | Deletion of the craniofacial-specific superenhancer distal to the Hoxa gene cluster mimics the Hoxa2 null phenotype. a** Schematic of deletion, mouse chr6:50673614-51196805 (mm10), spanning major predicted contacts with the *Hoxa* cluster. **b** (upper row) Three-dimensional rendered images generated from microCT scans of representative wildtype E18.5 embryos and their heterozygote and homozygote ΔGCR littermates. Ventral view of the skulls reveals multiple cranial base and palatal bony defects in homozygotes. The palatal defects include cleft palate in ~66% of homozygotes and reflected by marked separation of the palatine processes of the maxillae [ppm], separation and ventrally projecting palatine bones [pb], as well as lateral flattening of the medial pterygoid processes [mpt] of the basisphenoid. There was variability in the palatal presentation in homozygotes (see Figure S22 for all scanned embryos). The cranial base presentation, characterized by the notably abnormal appearance of the lateral pterygoids [lpt] of the basisphenoid and abnormal anterior shape of the basioccipital (denoted by the arrowhead), was fully penetrant in homozygotes. A posterior cleft (arrowhead) or small notch in the basisphenoid was also evident in ~50% of homozygotes. **(lower row)** Soft tissue rendering from microCT scans of E17.5 embryos confirm the cleft palate observed in some homozygotes. Note the normal formation of rugae despite the cleft. **c** Left lateral view of the bony (top) and soft tissue (bottom) rendering of microCT scans of littermates. Homozygotes show mirror duplications of the tympanic ring (tr; *tr), tympanic process and squamous bone (tp/sq; *tp/*sq) reminiscent of previously reported *Hoxa2* null mice. The abnormal lateral pterygoid (lpt) of the basisphenoid is evident from this view of homozygotes. Although not previously described in *Hoxa2* null mice, the mandibular angle was consistently hypoplastic (arrowhead) in homozygotes. On the soft tissue renderings, variable severity microtia can be clearly seen in homozygotes (arrow). Microtia ranges from grade I to grade III.

*Hoxa*^ΔGCR/ΔGCR^ mice compared to WT and *Hoxa*^+/ΔGCR^ littermates (Supplemental Table 12). MicroCT scans of *Hoxa*^ΔGCR/ΔGCR^ embryos compared to WT and *Hoxa*^+/ΔGCR^ littermates at E18.5 confirmed the incidence of secondary palatal clefting in homozygous embryos, evident both from renderings of the palatal shelf soft tissue and the marked separation of the palatine processes of the maxillae, separation and ventral projection of the palatine bones, and lateral flattening of the basal pterygoid processes of the basisphenoid (Supplemental Movie 1). Despite these underlying bony anomalies and the separation of the shelves, the palatal rugae pattern appeared largely normal. Notably, the *Hoxa*^ΔGCR/ΔGCR^ and *Hoxa*^+/ΔGCR^ embryos had hypoplastic mandibular angles, albeit more significant in homozygous embryos. In addition to the flattened basal pterygoids, the lateral pterygoids of the basisphenoid were also strikingly abnormal, with large lateral bony projections in *Hoxa*^ΔGCR/ΔGCR^ embryos (Figs. 6b **upper**, S18, Supplemental Movie 2, Supplemental Movie 3). Unlike the incomplete penetrance of cleft palate, the basisphenoid anomalies and an abnormal anterior basioccipital shape were fully penetrant in homozygotes. A posterior notch in the basisphenoid was also evident in ~50% of homozygotes. In *Hoxa*^ΔGCR/ΔGCR^ embryos with cleft palate, ventral projections from the palatal bones instead of the normal horizontal projection toward the midline suggested that the palatal shelves had not elevated; however, soft tissue rendering from the same scans showed relatively normal rugae formation despite the failure of the shelves to approximate and the aberrant underlying palatal bone projections. Most *Hoxa*^ΔGCR/ΔGCR^ embryos without cleft palate (5 of 6) exhibited overtly normal palatal bones, maxillary palatine processes and basal pterygoids (Fig. 6b, **lower**). Despite the variably penetrant clefting, all *Hoxa*^ΔGCR/ΔGCR^ embryos showed mirror-image duplication of the tympanic ring and potential partial mirror-image duplication of the tympanic process and squamosal bone (Fig. 6c **upper**; Supplemental Movie 4). Soft tissue rendering also showed that external ears (pinna) were overtly hypoplastic in all *Hoxa*^ΔGCR/ΔGCR^ embryos, although the severity varied from embryo to embryo. The bony and soft tissue ear phenotypes did not correlate with palatal clefting (Fig. 6c, **lower**). Collectively, the developmental phenotypes seen in the *Hoxa*^ΔGCR/ΔGCR^ mice were essentially identical to those reported in *Hoxa2*-/- lines[31,51,52] suggesting that the lack of the GCR had very specific consequences on craniofacial gene expression.

used to establish our breeding colony. The F1 mice were overtly normal and fertile. Since this was such a large genetic manipulation and to eliminate interference between potential off target effects, we performed multiple backcrosses utilizing C57B6 wild type mice and heterozygous GCR deletion (*Hoxa*^+/ΔGCR^) mice. The original F1 mice remained healthy during this time, and we obtained expected numbers of heterozygous offspring across all backcrossing. Upon crossing heterozygous *Hoxa*^ΔGCR/+^ mice, perinatal lethality was observed in all homozygous pups. Pups were grossly normal externally, but when we examined the internal craniofacial structures, we found 55% (*n* = 6/11) had clefts of the secondary palate. (Supplemental Table 12). We inspected embryos at multiple stages of development and did not observe any distinct morphological differences before E14.5. However, at this stage the palatal shelves frequently failed to fuse in the *Hoxa*^ΔGCR/ΔGCR^ embryos. Limb staging and comparison of fetal weight at E17.5 did not indicate significant body-wide developmental differences in

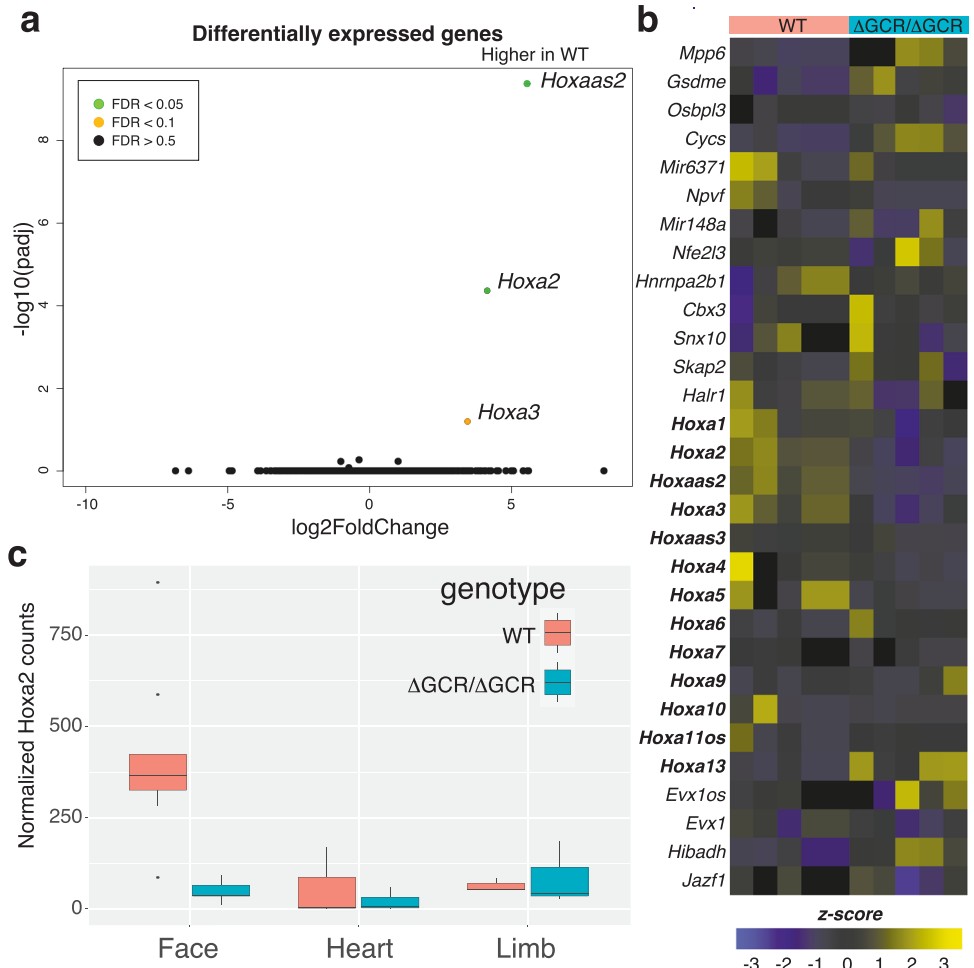

**Fig. 7 | Deletion of a superenhancer region has specific effect on *Hoxa* genes.**
**a** Deletion resulted in decrease in Hoxa gene expression without similar decrease in expression of intervening genes such as *Snx10* and *Skap2*. Significance values were calculated with Wald test by DESeq2 and adjust with the Benjamini-Hochberg approach. **b** Heatmap of expression from all replicates of WT and ΔGCR/ΔGCR for genes indicated in panel **a**. **c** *Hoxa2* expression was substantially different in E11.5

ΔGCR/ΔGCR vs. WT littermates in craniofacial tissue but not in heart or limb. Measurements are independent biological replicates: Face WT *n* = 7, ΔGCR/ΔGCR *n* = 5; Heart WT and ΔGCR/ΔGCR *n* = 3 each; Limb WT and ΔGCR/ΔGCR *n* = 3 each. The center line denotes the median value (50th percentile), the box contains the 25th to 75th percentiles and the whiskers mark the 5th and 95th percentiles. Data points beyond these values are shown as individual dots.

To determine if *Hoxa2* gene expression or additional genes and tissues were impacted, we collected multiple tissues (craniofacial, heart, and limb) from embryos at E11.5; a stage prior to the onset of phenotypic differences in *Hoxa*^ΔGCR/ΔGCR embryos and when *Hoxa2* gene expression is robust[48]. Comparison of global gene expression of craniofacial tissue between *Hoxa*^ΔGCR/ΔGCR and wildtype type embryos revealed specific effects on *Hoxa2*, *Hoxa3*, and *Hoxaas2* (Fig. 7a). We did not observe any significant changes in expression for other genes surrounding the deletion nor residing between the deletion and the *Hoxa* gene cluster (Fig. 7b). We also examined expression of genes important for limb and heart development as there were limb and heart superenhancers nearby (Figure S19), with both limb and craniofacial enhancer modules enriched for transcription factor binding motifs for Pbx1 and Pitx1 (Figure S20). When we examined heart and limb tissue, we found a small number of differentially expressed genes between *Hoxa*^ΔGCR/ΔGCR and wild type mice, however none of these were at this locus, the *Hoxa* gene cluster maintained consistent expression across all embryos (Figure S21), and limb morphology was not altered in *Hoxa*^ΔGCR/ΔGCR (Figure S22). Inspection of the three-dimensional architecture of this region using HiC in craniofacial tissue from *Hoxa*^ΔGCR/ΔGCR mice confirmed complete deletion of the TAD subdomain containing the GCR (Figure S23). This resulted in a smaller TAD domain that excluded the anterior half of the *Hoxa* gene cluster but

otherwise did not cause further enhancer adoption or large-scale changes in chromosome conformation. These findings were consistent with the very specific effect on expression of the *Hoxa* gene cluster and none of the other nearby genes. Overall, these data support the conclusion that deletion of an entirely non-coding craniofacial-specific superenhancer region has tissue-specific consequences on a small number of genes located a considerable distance away. Mice completely lacking this region frequently had craniofacial abnormalities and died soon after birth.

### Identification of novel copy number variations affecting the GCR and associated with craniofacial abnormalities in humans

Given the strong phenotype in the mouse homozygous knockout, we wondered if similar phenotypes might be apparent in human development. To address this, we first examined a curated database of patients with a variety of developmental abnormalities likely due to copy number variation. Of the 21 Individuals within the DECIPHER Database (deciphergenomics.org[53]) with copy number variants (CNVs) overlapping the region (chr7:25,580,400-25,849,400;hg19), 14 of these had reported phenotypes, 10 of which had some type of craniofacial abnormality, a significantly higher incidence compared to the DECIPHER database as a whole (Figures S24 and S25, **and** Supplemental Table 13). When we examined a large collection of whole

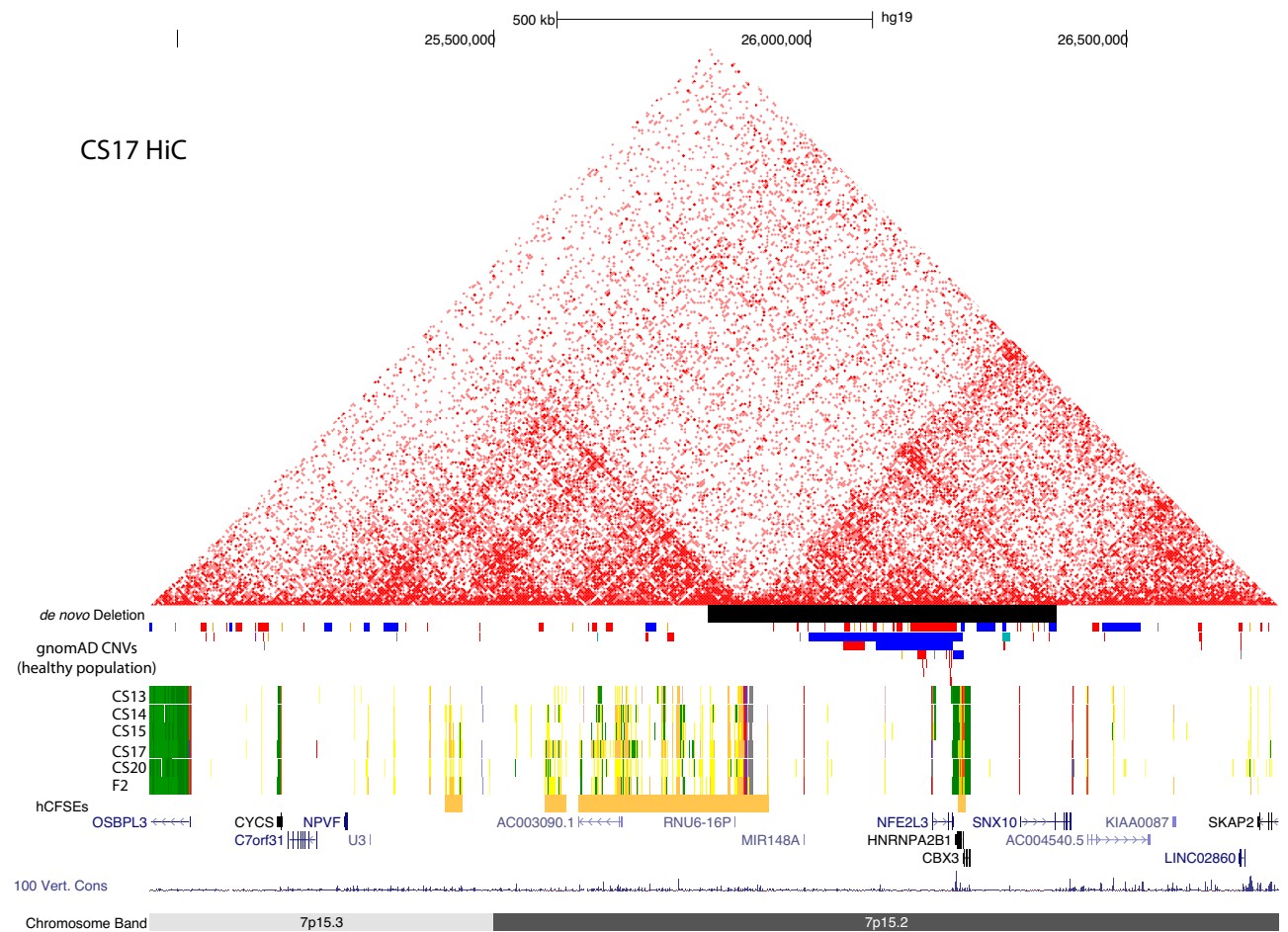

**Fig. 8 | Location of de novo deletion overlapping GCR.** Browser image (hg19) showing de novo deletion presented by the group at Ghent University corresponding to Chr7:25,839,621-26,389,620 on the hg19 assembly. Also shown are the UCSC Browser tracks for the 25-state model, human embryonic craniofacial superenhancers and gnomAD Structural Variation track filtered to show CNVs >300 bp. Colors in the gnomAD track are as they appear in the UCSC genome browser, red bars signify deletions and blue bars duplications.

genome sequences from otherwise healthy adults, we did not observe any large copy number variations (>25 kb) overlapping the GCR or the *HOXA* gene cluster (Figure S23[54]). Copy number variants were observed that encompassed *NFE2L3*, *HNRNPA2B1*, and *CBX3* but none extended across the 3′TAD boundary of the GCR (Fig. 8). Together these findings strongly suggest that this GCR is important for human craniofacial development and associated with craniofacial abnormalities when copy number is altered. However, all the copy number changes identified in DECIPHER were quite large (>3MB) and included many genes outside of the noncoding region of interest. Thus, it remained unclear if alteration of this region alone could result in craniofacial abnormalities in humans.

During the course of this work, we identified two novel copy number changes that were much smaller than those reported in DECIPHER, but also associated with severe human craniofacial abnormalities. The first case was a fetus with severe craniofacial abnormalities identified at Ghent University Hospital. Following targeted ultrasonography (see gynecologists report in Supplemental Text), the pregnancy was terminated at 14 weeks. The autopsy revealed bilateral cleft lip and palate, an underdeveloped nose with a single nostril, as well as clubfeet and anal atresia (Figure S26). Shallow whole genome sequencing identified a 550 kb deletion (chr7:25800001-26350000, hg38), which was absent in the parents and encompasses the *NFE2L3*, *HNRNPA2B1*, *CBX3*, *MIR148A* genes, as well as exons 1 and 2 of the *SNX10* gene (Figure S27). *SNX10* has been associated with autosomal recessive osteopetrosis which includes clinical features

such as macrocephaly and facial anomalies (OMIM #615085); however, it is not a highly constrained gene in the gnomAD database[54] (pLI = 0, LOEUF = 1.07) suggesting loss of a single copy is well tolerated in humans. Targeted analysis of over 4000 genes implicated in monogenic diseases was performed by the Center for Medical Genetics at Ghent University Hospital for this trio (see Methods) but did not reveal any other putatively causal variants. The deletion overlapped the 3′ end of the *HOXA* GCR we identified, including the 3′ TAD boundary, which forms strong interactions with the *HOXA* gene cluster, and the validated HACNS50 enhancer element. While supportive of our hypothesis, no other material was available for this case preventing further study of tissues from this sample or generation of iPSCs.

The second case was a two year old male patient born with multiple congenital anomalies, primarily affecting craniofacial structures identified at Cincinnati Children's Hospital Medical Center (CCHMC) (Fig. 9a). Postnatal exam demonstrated hypertelorism, low-set and posteriorly rotated ears, bilateral ablepharon with hair tufts at lateral corners of the eyes. He had 10 rib pairs and a midline pseudo cleft lip. Brain MRI demonstrated marked decreased volume of the cerebellum and associated decreased size of the pons (Fig. 9b). The cerebellum had irregular morphology with a disorganized foliar pattern. Initial CT scan demonstrated partial acrania involving the frontal calvarium and anterior skull base, bilateral choanal atresia/stenosis, micrognathia with glossoptosis and midline cleft of secondary palate. The inner ears were also abnormal, with incomplete formation of bilateral semicircular canals and suspected bilateral oval window atresia or stenosis.

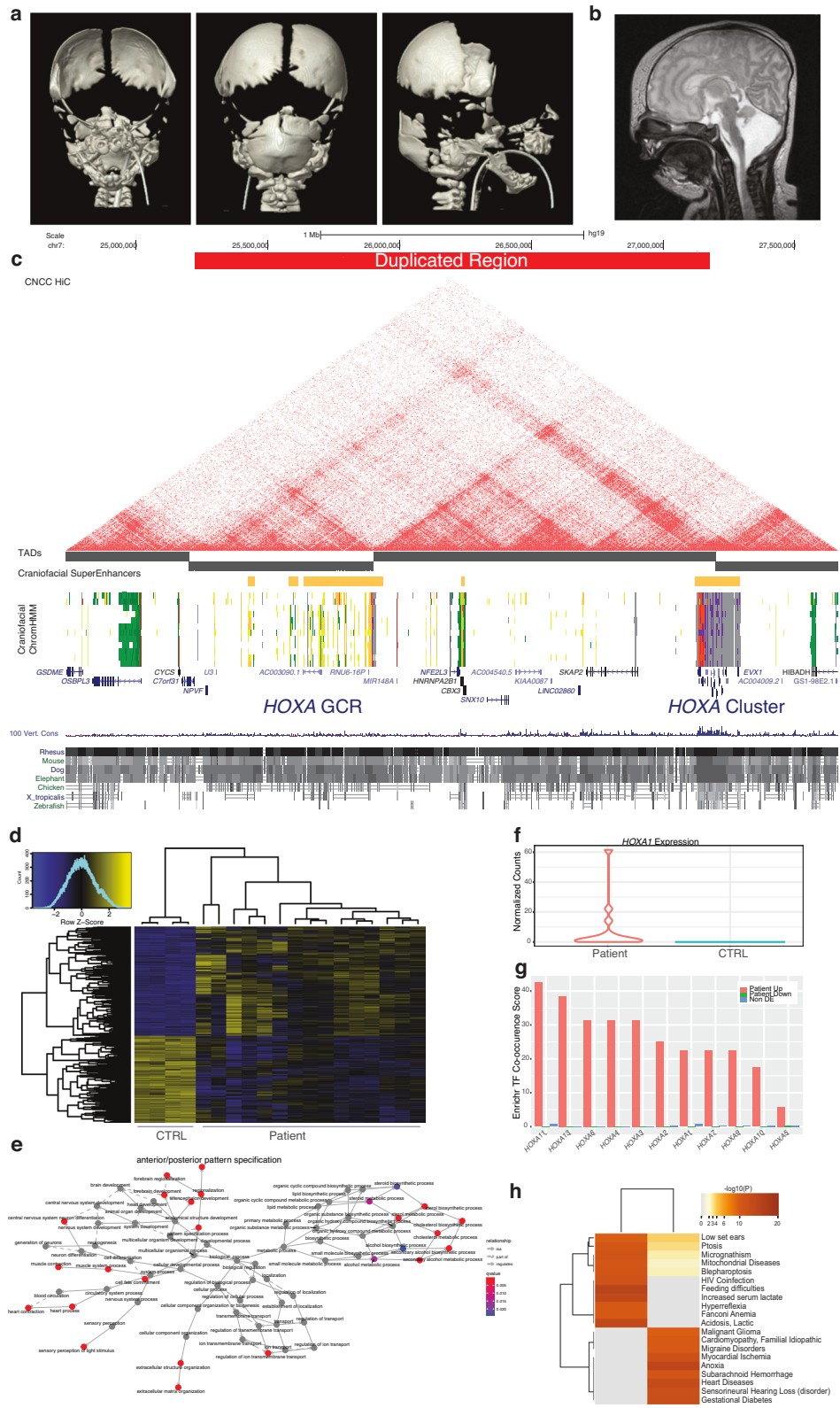

**Fig. 9 | *HOXA* GCR Duplication Patient. a** Newborn 3D CT with reconstruction demonstrating partial calvaria. **b** Newborn brain MRI demonstrating hypoplastic cerebellum and brainstem. **c** UCSC Genome Browser shot showing duplicated region identified in this patient. **d** Differentially expressed genes across multiple replicates (*n* = 3) and clones (*n* = 5) of patient neural crest cells compared to neural crest derived from control cell lines (*n* = 4) (CTRL). **e** Network plot of enriched gene ontology terms of all significantly dysregulated genes between WT and patient neural crest. **f** Normalized counts from all patient-derived NCC RNA-Seq. **g** Transcription factor co-occurrence scores reported by Enrichr for upregulated (red), downregulated (green), and non-differentially expressed genes (blue) for each of the *HOXA* TFs. **h** Significant disease ontology categories identified by Fisher Exact Test (one-sided, Benjamini-Hochberg adjusted) calculated by DisGeNet for down and upregulated genes in patient neural crest.

Follow up CT scans at 18 months of age confirmed multiple anomalies that coincided with frontonasal dysplasia, a complete absence of the frontal bones and hypoplasia of the anterior parietal bones (Figure S28).

Commercial exome trio testing (GeneDx, Gaithersburg MD) identified a maternally inherited variant of uncertain significance in *SNRPB*: c.530 C > A, p.(Pro177Gln). Clinical gene-specific RNA sequencing on blood (MNG Laboratories) did not identify changes in transcript levels or splicing patterns for *SNRBP*. The only known mechanisms of pathogenicity for *SNRPB* are variants leading to abnormal splicing and premature stop codon (PMID 25047197). Since the patient's mother had no craniofacial anomalies, this variant was not pursued relative to the patient's phenotype. Subsequent microarray testing at CCHMC identified 2 duplications of uncertain significance: DGV(GRCh37:Feb.2009) (hg19): arr [GRCh37] 7p15.3p15.2(25259224_27193908)x3, and 11q24.1(123472528_123505099)x3. Due to the 7p duplication encompassing the *HOXA* cluster and its involvement in craniofacial development as described throughout this work, this duplication was further investigated.

Targeted long-read sequencing confirmed that the 7p duplication was indeed tandem. Visualization of aligned reads revealed that the exact breakpoints of the duplication were at chr7:25,220,918 (hg38) and chr7:27,176,187 (hg38), the latter of which lies within a low-complexity GA-rich region within *HOXA10* (Figure S28). The duplicated region contained complete copies of *HOXA1, HOXA2, HOXA3, HOXA-AS2, HOXA4, HOXA5, HOXA6, HOXA7,* and *HOXA9*. The second copy of HOXA10 was bisected by the duplication breakpoints. This duplication places the *HOXA* GCR directly adjacent to *HOXA* genes 1 through 9 (Fig. 9c).

To determine if this duplication was causative for the patient's phenotypes, multiple clones of induced pluripotent stem cells were generated. Patient and control iPSCs were differentiated into NCCs and profiled for gene expression. Over 6000 genes were differentially expressed in the patient iPSCs relative to control cells (Fig. 9d, Supplemental Table 15). Among differentially expressed genes there was significant enrichment for those associated with anterior-posterior pattern specification, brain and eye development, extracellular matrix organization, and cholesterol biosynthesis (Fig. 9e). Notably, significant upregulation of *HOXA1* expression was observed in patient iPSCs (Fig. 9f). While binding profiles of most of the *HOXA* genes have not been investigated in tissues or cell types relevant to craniofacial development, we wondered whether other gene expression datasets contained evidence for the direct regulation of these genes by *HOXA* transcription factors. We leveraged over 300,000 submissions to the Enrichr gene set enrichment web portal to assess the co-occurrence of genes with over 800 TFs[55]. This approach has been shown to accurately capture known gene-gene interactions and discover novel ones[56]. When upregulated and downregulated genes were independently analyzed with this approach, we found that upregulated genes had significantly higher co-occurrence scores with all *HOXA* TFs (Fig. 9g). Down regulated genes were only significant for one TF, *HOXA5*, but at a much lower co-occurrence score. Random permutation analysis of 3403 genes that were not differentially expressed found no significant co-occurrence with any *HOXA* TFs. Given this discrepancy between up and downregulated genes, we investigated these genes independently for disease related phenotypes. Enrichment for upregulated genes was associated with many phenotypes observed in this patient including low-set ears and micrognathia (Fig. 9h). Enrichment for downregulated genes was associated with genes related to hearing loss, which agrees with the abnormalities observed of the inner ears. Interestingly, most of the phenotypes that were significantly enriched among downregulated genes were related to heart function, though cardiac abnormalities were not observed in this patient. Overall, our findings strongly support that this copy number variant

resulted in improper regulation of *HOXA* gene expression and demonstrated the importance of the GCR we have identified for normal human development.

## Discussion

Collections of coactivated enhancers commonly referred to as super-enhancers encompass genes that play important roles in tissue and disease specific biology. Our identification and global analysis of superenhancers active in craniofacial development largely confirmed these findings and demonstrated that many craniofacial related transcription factors are likely controlled by such regions. Moreover, we found that superenhancers specific to craniofacial development are enriched for transcription factors and other genes that are directly linked to craniofacial related diseases. These included *ALX4, MSX2,* and CYP26 family members, which demonstrated clear craniofacial-specific regulatory landscapes. Interestingly, many craniofacial-specific superenhancers we identified neither included, nor were in close proximity to known craniofacial genes.

We reasoned that such regions might either contain a previously unannotated gene or operate over long distances to target a craniofacial relevant gene. Unlike individual enhancers, the regions we identified occupy fairly large portions of the genome, making them challenging to study as a complete unit. Furthermore, deletion of many individual enhancers and even megabase-scale, completely noncoding regions have resulted in mice with no overt phenotype[15,18,20,37,38]. These findings call into question the role such regions might play during development. However, we observed functional gene enrichments in the surrounding areas that suggest an important role for these regions in craniofacial development and disease.

To address this, we focused on completely noncoding super-enhancers whose removal was not predicted to disrupt a gene. We prioritized regions by overlaying multiple layers of functional genomics data and publicly available enhancer validation experiments and tested one of the highest priority regions. This region had nearly one hundred active craniofacial enhancer segments, several VISTA validated craniofacial enhancers, showed functional conservation in development between both human and mouse, and was embedded in a large syntenic region that also included the *HOXA* gene cluster. Examination of chromosome conformation data revealed that this region formed a distinct topologically associated domain (TAD) in most tissues. Interestingly, in craniofacial tissue this region becomes a sub-TAD of a larger domain that includes the *HOXA* gene cluster.

TAD organization has been shown to be important for gene regulation. Breaking or inverting TAD boundaries to affect their enhancer-gene content can result in developmental abnormalities[41,57–59]. In an attempt to separate the function of the novel superenhancer region from larger scale TAD organization, we attempted to create a "scarless" edit with respect to TAD boundaries. We targeted strong CTCF binding sites at both TAD boundaries in human cells. While we were unable to create homozygous deletions in human embryonic stem cells, we identified a cell line that lacked one copy of the region and harbored a ~600 kb inversion of the remaining copy. This inversion cell line showed that this region functioned as an enhancer in the classical sense, regulating gene expression over long distances independent of orientation. NCC differentiation of this inversion line did not reveal significant differences in gene expression of the *HOXA* gene cluster, nor any other genes in the direct vicinity.

Interestingly, we observed an increased frequency of interactions between the TAD boundary that was now 600 kb further away from the *HOXA* gene cluster and the anterior *HOXA* genes. This suggested that this boundary in particular was programmed to target the *HOXA* gene cluster in NCCs and functioned over even greater distances, even dominating over other boundary regions that are closer to the target and have properly oriented CTCF motifs for looping[43,60,61]. Our allele specific analysis confirmed the HiC findings through biased expression

from the HOXA cluster that was in *cis* with the inverted regulatory region. While the lack of impact on gene expression in NCCs was unexpected, our inability to identify homozygous knock-outs in H9 ESCs and the behavior of the inversion cell line suggested that the absence of this region might have a strong impact on *HOXA* gene expression and was essential for human stem cell survival. Recent work in hematopoietic stem and progenitor cells (HSPCs) support this conclusion[47], where a 17 kb deletion encompassing the strong TAD boundary more proximal to the *HOXA* gene cluster led to an increase in differentiated cells. This suggested a role for the interaction between the anchor and the *HOXA* cluster in the maintenance of stem cell identity. Disruption of this region, identified in HSPCs as a DNA methylation canyon, led to altered expression of *HOXA* genes without significant changes in expression for the intervening genes. These results along with our findings point to this region being a potential global control region of the *HOXA* gene cluster, similar to those already known to control other HOX gene clusters[36,62,63].

Our experiments in cell culture, while suggestive, failed to address the original question of whether specific superenhancers are important for development. We therefore generated an orthologous deletion in mouse. While the heterozygous founders and many of their resulting progeny lacked any overt phenotype, homozygous mice never survived postnatally beyond 48 hours and frequently had clefts of the secondary palate. Gene expression analysis of craniofacial tissues at E11.5, prior to the emergence of clefting phenotypes, showed specific effects on the anterior *Hoxa* genes, particularly *Hoxa2*, and no other genes in the region. Other tissues from the same stage of development showed no such issue in gene expression. Given the strong effects on *Hoxa2* gene expression in our mice and the documented role of *HOXA2* in craniofacial development, we more closely examined the skeleton of homozygous deletion mice using microCT. We found several phenotypes within the craniofacial skeleton including dysmorphic basisphenoid bone, tympanic process and tympanic ring, as well as microtia in the $Hoxa^{\Delta GCR/\Delta GCR}$ mice we generated. This presentation was strikingly similar to those seen in previously published *Hoxa2* null mice[31,51,52,64] and to recently reported somewhat smaller deletions of this same region[65]. In particular, Kessler and colleagues subdivided this larger gene desert into two regions they referred to as *Hoxa* Inter-TAD Regulatory Elements 1 and 2 (HIRE1 and HIRE2). HIRE2 was orthologous to the smallest craniofacial superenhancer we identified in this gene desert (39 kb) and overlapped a superenhancer call from the developing human heart (Fig. 2a). Deletion of this region was reported to only result in microtia. HIRE1 was over 175 kb in size and encompassed two superenhancers we identified in Fig. 2b. This deletion showed very similar phenotypes to the large deletion we describe here. However, this deletion did not include the region that is a superenhancer in many other tissues, the HACNS50 enhancer, or the bivalent chromatin domain that likely indicates the TAD boundary of this region. Notably the HIRE1 deletion also excluded regions affected by the de novo deletion identified in this work and depleted of copy number variation in gnomAD (Figures S23b and S26c). These results combined with ours here define the critical region that can cause such phenotypes in mouse, but dozens of conserved putative enhancer sequences remain in this interval. Further dissection of the *HOXA* GCR is thus required to determine whether there are one or more smaller enhancer elements primarily responsible for the regulatory activity and phenotypes in each species. Given the specific distribution and functions of the anterior *HOXA* genes, it will be important to determine whether portions of the superenhancer have distinct control over expression of specific *HOXA* genes. Deletion of select elements could impact the limbs or other organs despite not being affected by deletion of the entire superenhancer. CRISPR-Cas9 directed dissection of the superenhancer cluster coupled with single-cell ATAC-seq could provide a means to identify whether there are developmental- and/or cell type-specific modules within the *HOXA*

GCR. This region could also serve as an important locus to dissect the controversy surrounding whether individual enhancers are linearly additive in their activity or if more complex activities arise from these individual components ([11], Shin et al. 2016, Dukler et al. 2016,[13,15], Zuin et al. 2022).

These results in mice, along with the human fetus and patient we described above, point to a novel dosage controlling function of this GCR for craniofacial *HOXA* gene expression that is essential for life in both humans and mice. Within the region of the human deletion, the only genes with comparable constraint to anterior *HOXA* genes are *CBX3* and *HNRNPA2B1* (LOEUF score 0.67, 0.22 respectively). However, haploinsufficiency of either gene does not produce phenotypes similar to the described fetus[66–69]. The other distinct difference between humans and mice is that loss of a single allele of the GCR in mice only yielded subtle cranial skeletal features in a few individuals, whereas perturbing a single allele in humans can potentially lead to catastrophic effects. As we noted above, a sequence with a high number of human-specific substitutions, HACNS50, is located in close proximity to the TAD boundary of the GCR[28]. This could indicate that this region has gained or lost activities specifically in humans, resulting in human-specific sensitivity to loss of this region. The newly reported tandem duplication patient also indicates increased dosage of this enhancer in the context of *HOXA* gene regulation is also detrimental. Overexpression of *Hoxa2* by approximately 60% in mouse CNCCs has been reported to cause craniofacial abnormalities strikingly similar to those observed in this patient, including hypoplastic mandible and frontal skull bones[70]. In closing, we have shown compelling evidence that copy number changes in this region in human are deleterious and should no longer be considered variants of unknown significance.

## Methods

### Mouse and human ethics statement

All protocols for animal use and care were approved by the University of Connecticut Institutional Animal Care and Use Committee (Protocol AP-2000061-0723) and conform to the NIH Guide for the Care and Use of Laboratory Animals. $CO_2$ euthanasia was performed in accordance with the AVMA Guidelines for the Euthanasia of Animals. The use of human embryonic tissue was reviewed and approved by the Human Subjects Protection Program at UConn Health (UCHC 710-2-13-14-03). Human embryonic craniofacial tissues were collected via the Joint MRC/Wellcome Trust Human Developmental Biology Resource (HDBR) under-informed ethical consent with Research Tissue Bank ethical approval (18/LO/0822 and 18/NE/0290, project 200225). Donations of tissue to HDBR are made entirely voluntarily by women undergoing termination of pregnancy. Donors are asked to give explicit written consent for the fetal material to be collected, and only after they have been counseled about the termination of their pregnancy. Further documentation of all policies and ethical approvals for HDBR sample collection can be found at https://www.hdbr.org/ethical-approvals. Tissues were flash-frozen upon collection and stored at −80 °C. Upon thawing, the samples were quickly inspected for intactness of the general craniofacial prominences and processed for HiC as described below.

### Mouse husbandry, tissue collection and imaging of palates

Male and female heterozygous for deletion of chr6:50,673,614-51,196,805 (ΔGCR/+) on C57BL/6 background were provided by Cyagen Animal Model Services (Santa Clara, CA, USA). Founders were backcrossed to WT C57BL/6J mice ordered directly from Jackson Laboratories (Bar Harbor, ME, USA) and four generations of backcrossing ΔGCR/+ males to WT females was performed to reduce phenotypes due to off-target CRISPR effects. Mice were housed according to recommendations by Jackson Laboratory with 12 h light:dark cycle beginning at 7 a.m. The ambient temperature was maintained between

20 and 22 °C and humidity was maintained at 40–60%. Mice were given ad libitum access to food and water. Timed matings were established by the identification of vaginal plugs the morning following the housing of a single male with multiple female mice. Pharyngeal arches from E11.5 embryos were collected as follows: timed matings were set up and the mice separated the next day and females checked for plugs. Noon of that day was counted as day 0.5. On day 11.5, mice with noted plugs were euthanized with $CO_2$ in accordance with the AVMA Guidelines for the Euthanasia of Animals. Uterine horns were removed from the euthanized female and rinsed in PBS, individual embryos were removed from the uterine sacs and individually dissected for pharyngeal arches, limb buds, heart and forebrain. The dissected tissue was frozen on a dry ice/ethanol slurry and stored at −80 °C until use. The remainder of the body was used for genotyping. DNA for genotyping was extracted using a high salt lysis buffer with proteinase K followed by chloroform extraction. Primers spanning the deletion were used to confirm genotype (sequences in Supplemental Table 11). For initial assessment of palatal morphology and fetal weight, timed matings were set up between ΔGCR/+ mice. The mice were separated the following morning and noon of that day was counted as day 0.5. At day 17.5, the female mice were checked for pregnancy and euthanized with $CO_2$ in accordance with the AVMA Guidelines for the Euthanasia of Animals. The uterine horns were removed from the euthanized mother, rinsed in cold PBS, the individual pups removed from the uterine sacs, rinsed in PBS, gently dried and weighed, a tail clip taken for genotyping and the body fixed in neutral buffered formalin for at least 48 h prior to removal of upper jaw for palate photographs. For soft tissue and skeletal imaging, timed matings were similarly set up and embryos taken at E18.5 as described above. Embryos were fixed in neutral buffered formalin for 24 hours and transferred to 70% ethanol prior to shipping for analysis.

## ROSE algorithm

We used the Rank Ordering of Super-Enhancers (ROSE) algorithm[6,71]. Briefly, H3K27ac ChIP-seq and Input (control) data and genomic regions classified with active chromatin states were provided to the algorithm. For the 25-state model, these states included 1_TssA, 2_PromU. 3_PromD1, 4_PromD2, 9_TxReg, 10_TxEnh5', 11_TxEnh3', 12_TxEnhW, 13_EnhA1, 14_EnhA2, 15_EnhAF, 16_EnhW1, 17_EnhW2, 18_EnhAc, 19_DNase, 22_PromP, and 23_PromBiv. For the 18-state model, these states included 1_TssA, 2_TssFlnk, 3_TssFlnkU, 4_TssFlnkD, 7_EnhG1, 8_EnhG2, 9_EnhA1, 10_EnhA2, 11_EnhWk, 14_TssBiv and 15_EnhBiv. Enhancers within 12.5 kb were "stitched" together as a single region. "Stitched" enhancers as well as enhancers without another enhancer within 12.5 kb were ranked for H3K27ac enrichment. Determination of superenhancer was based on the threshold of where the slope of H3K27ac enrichment = 1.

## Transcription factor binding motif identification

Superenhancers were determined for mouse tissues using the ROSE algorithm as described above provided with data from mouse ENCODE (ref) and two mouse E11.5 craniofacial tissue samples assayed in our lab. Enriched transcription factor motifs were identified for mouse E11.5 craniofacial, E11.5 heart and E11.5 limb strong enhancer segments (states 8, 9 and 10) found in superenhancers located in the region chr6:50673614-51196805 (mm10). HOMER findMotifsGenome (ref) was applied to the strong enhancer segments using a background of repressed and quiescent chromatin states (states 17 and 18) found within the region covered. Known motifs with enrichment $p$-value ≤ 0.01 were reported by HOMER.

## Gene ontology and disease ontology

Biological Process Gene Ontology and Disease Ontology from DisGeNet enrichment was primarily performed with the R packages clusterProfiler (3.14.3) and DOSE (3.12.0)[72]. Genes overlapping TSSs

or assignment of nearest genes was performed using BEDTools (2.29.0)[73]. LiftOver analysis was performed using KentUtils (1.04.00). In some cases Disease Ontology enrichment was identified through WEB-based GEne SeT AnaLysis Toolkit (WebGestalt) ([74,75]; webgestalt.org).

## In-vivo validation of HACNS50

In vitro lacZ reporter assay of an additional enhancer element was performed by Justin Cotney. A 2.6 kb segment centered on the conserved sequence corresponding to HACNS50[28] was amplified from human genomic DNA by polymerase chain reaction (PCR) using the following primers: HACNS50 F 5'-CACCCCATTTCTGAGGGGGAAATAA-3', HACNS50 R 5'- TTATTTCCTTCAGGCCCTTG-3', and cloned into an Hsp68-lacZ reporter vector as previously described[76]. Generation of transgenic mice at the Yale University Transgenic Mouse Facility and embryo staining were carried out as previously described[76]. We required reporter gene expression in a given structure to be present in at least three independent transgenic embryos as assessed by two researchers to be considered reproducible.

## H9 hESC cell culture

Routine culture of H9 cells was done in feeder-free conditions using Matrigel substrate (Corning, 354277) and Essential 8 media (Gibco, A1517001). Where feeder cells were used (following nucleofection with gRNAs) the cells were plated on DR4 MEFs in a gelatinized 10 cm tissue culture dish with hESC on MEF media (DMEM high glucose + 10% FBS).

## Guide RNA design and preparation

We used the online tool at http://crispor.tefor.net/ to select appropriate gRNAs. Guide RNA oligonucleotide pairs were synthesized by IDT (Coralville, IA, USA) and received in lyophilized form. The gRNA oligos were diluted to 100 mM with sterile water. The oligonucleotide pairs for each gRNA were phosphorylated and annealed in a reaction containing 100 mM of each oligonucleotide, 5U T4 PNK (NEB, M0201) and T4 ligation buffer at 37 °C for 30 minutes with boiling at 95 °C for 5 minutes followed by a gradual cool to 25 °C at an approximate rate of 5 °C per minute. The resulting annealed oligos were diluted 1:200 and added to a ligation reaction with 50 ng BbsI digested pX459 plasmid, 1x Quick Ligation buffer and Quick Ligase (NEB, M2200). The ligation reaction proceeded for 10 minutes at room temperature and then was treated with PlasmidSafe Exonuclease. The Exonuclease reaction contained the full volume of the ligation reaction, 1X PlasmidSafe Buffer, 1 mM ATP, and 3.2U PlasmidSafe Exonuclease (Epicentre, E3101K) and was incubated at 37 °C for 30 minutes. The recombinant plasmids were introduced into competent dh5alpha cells via heat shock at 42 °C, recovered in SOC at 37 °C for 60 minutes with gentle rocking and plated on LB-Ampicillin plates. After overnight incubation of the plates at 37 °C in a bacterial incubator, colonies were selected and grown overnight in 3 ml liquid LB-Ampicillin at 37 °C in a shaking incubator. Glycerol stocks were prepared from a fraction of each culture. Plasmid DNA was extracted from the overnight cultures using Qiagen miniprep kit (Qiagen, 27106) according to the manufacturer's instructions for centrifuge column prep. Quantity of DNA was measured using Nanodrop spectrophotometer and proper construction of plasmid was confirmed using an EcoRI/BbsI double digest. Plasmid preps with the proper restriction digest profile were sent to Genewiz (South Plainfield, NJ, USA) for sequence confirmation by Sanger sequencing. The plasmid DNA was sent pre-mixed with LKO.1 5'_U6primer (5'-GACTATCATA TGCTTACCGT-3'). Glycerol stocks of the plasmids that were confirmed correct by Sanger sequencing were grown and prepared for maxiprep plasmid isolation (Zyppy Plasmid Maxiprep Kit, Zymo 6431). Proper cloning was achieved for all gRNAs.

## Genome editing of H9 hESCs

Guide RNAs gRNA1 and gRNA6, which together would cleave out the region chr7:25,295,587-25,921,144 (hg19) were introduced into H9 hESCs (WiCell) by nucleofection (Lonza, Basel, Switzerland). To start, hESCs in one confluent well of a 6-well plate were singlized, first detached from the Matrigel plate with Accutase (Thermo Fisher, 00-4555-56) at 37 °C for 10 minutes, and pipetted to single cell suspension in Essential 8 media + 10 mM ROCK inhibitor (Tocris Bioscience, Y-27632 dihydrochloride). Approximately half of the cells were transferred to a 15 ml conical tube and centrifuged at 200 x *g* for 5 minutes. The media was completely removed and replaced by 100 µL of Nucleofection Mix (82 µL Nucelofector solution, 18 µL supplement, 2.5 µg gRNA1, 2.5 µg gRNA6. The cells were resuspended in the Nucleofection Mix with a p200 pipette tip by pipetting gently three times. The sample was transferred to the cuvette using the 2 ml pipette provided with the Nucleofection kit. The cells were placed in the Nucleofector and run on the program for hESC, P3 primary cell protocol. Following the time in the machine, 500 µL hESC on MEF media plus ROCK inhibitor was added to the cuvette. The cells were plated on 10 cm plates with Matrigel or DR4 MEFs. The cells were fed with the appropriate media plus ROCK inhibitor until colonies were visible. Selection with puromycin and support with ROCK inhibitor began the day following Nucleofection and continued for a second day. Subsequently only ROCK inhibitor was added to the media until colonies were visible. Colonies were screened via the hotshot method. Briefly, portions of colonies were picked and transferred with less than 200 µL media to a PCR tube within a strip of 8 tubes. The samples were centrifuged for 2 minutes or longer in a microfuge with PCR strip tube adapter. The media was carefully aspirated and the cells resuspended in 20 µL lysis buffer (25 mM NaOH, 200 mM EDTA). The cells were incubated for 45 minutes at 96 °C in a thermal cycler. Following the lysis, freshly made neutralization buffer (40 mM Tris, pH 5.0) was added to each tube and 5 µL of DNA was used as a template for the screening PCR reaction using primers. Clones identified as edited had the target ends amplified by PCR with high-fidelity Taq polymerase (Pfusion HF; NEB, M030) and sent for Sanger sequencing (Genewiz, South Plainfield, NJ, USA). Sequences of gRNAs and screening primers are in Supplemental Table 16.

## Neural crest differentiation from hESCs

Starting with one confluent well of a 6-well plate of H9 hESCs (WiCell) on Matrigel, the cells were detached using Accutase at 37 °C for 10 minutes and the cells pipetted into a single cell suspension using a 2 ml serological pipette and resuspended into a total volume of 10 ml of Essential 8 media. The cells were then pelleted by centrifugation for 3 minutes at 1000 x *g*. The supernatant was aspirated and replaced with NCC Media (DMEM/F12 (Gibco, 10565-018) plus B27 supplement (Gibco, 17504-044 and Penicillin/streptomycin (Life Technologies, 15140122) with 3uM Chiron (CHIR-99021; Tocris Bioscience) and 10uM ROCK inhibitor. The cells were resuspended by gentle pipetting with a serological pipette and passed through a 40 micron filter. The cells were counted with a hemocytometer and diluted to the desired density (30,000 cells per cm²) using the NCC Media with Chiron and ROCK inhibitor. Media was changed daily with NCC Media plus Chiron and ROCK inhibitor on the day following plating and NCC Media plus Chiron only for the remaining days. Differentiation is complete by day 5.

## Detection of allele-specific expression

Input and ChIP-seq data using antibodies against H3Kme1, H3Kme2, H3Kme3, H3K36me3, H3K9me3, H3K27me3, H3K27ac from H9-derived NCC ([29]; GSE197513) were used to form low-coverage genomic sequence, from which GATK HaplotypeCaller[77] was used to generate a variant call format file. Chromosome 7 was phased individually using BEAGLE 5.4[78], in combination with 50 samples from 1000

Genomes Project[79]. Allele-specific expression from RNA-seq data was detected using PhASER and PhASER_ae[80].

## Human primary tissue source

Pharyngeal arch and craniofacial tissue from unfixed, unsectioned embryonic tissue was obtained from the Human Developmental Biology Resource (HDBR), a part of the Wellcome Trust, in the UK.

## Human primary tissue fixation for HiC

Prior to beginning, all appropriate precautions were taken for the handling of potentially hazardous biological material. Individuals handling the samples wore disposable isolation gowns, disposable hair nets, disposable plastic face shields, disposable shoe covers and double layers of examination gloves. Surfaces and instruments involved in the processing of the samples were disinfected with 10% bleach afterwards. Samples were removed from −80 storage and removed from the tube by thawing in a small amount of PBS. The tissue was transferred to a dish containing cold PBS on a chilled microscope stage. The tissue was documented by photography through the microscope at several angles and photos of the tube were taken as well. The tissue was assessed as to whether it needed to be further dissected to isolate the specific tissue of interest and only the tissue of interest was transferred to a tube with 1 mL PBS. The tissue was homogenized with an electronic tissue disruptor (Polytron PT 1200E, Kinematica, Fisher Scientific, USA). For HiC, formaldehyde was added to the remaining volume of homogenized tissue to a final concentration of 1%, incubated at room temperature for 15 minutes with rotation, quenched with 1.5 M glycine added to a final concentration of 150 mM and incubated for 10 minutes at room temperature with rotation. The cells were pelleted by centrifugation at 2500 x *g* for 5 minutes at 4 °C. The supernatant was removed and discarded as formaldehyde waste and washed once with 1 mL cold PBS. The cells were pelleted again by centrifugation as before, the supernatant removed and the fixed pellet frozen in a dry ice/ethanol slurry. The fixed samples and samples in Qiazol were stored at −80C until time of use.

## HiC

Crosslinked cell pellets or crosslinked embryonic tissue was resuspended in 1x Cell lysis buffer as used for 4 °C and nuclei released in a prechilled dounce homogenizer by 10 strokes with a loose glass pestle, 20 minutes rest on ice followed by 40 strokes with a tight glass pestle as for 4 °C. The nuclei were pelleted by centrifugation at 2500 x *g* at 4 °C for 5 minutes, washed twice with 1x NEBuffer 3.1 and pelleted again as before. The nuclei were resuspended in 1x NEBuffer 3.1 and permeabilized by the addition of SDS to 0.1% SDS final concentration, incubated for 10 minutes at 65 °C with shaking at 800 rpm on a ThermoMixer. The SDS was neutralized with Triton X-100 to 1% final concentration and incubated at 37 °C for 10 minutes with gentle rocking. Additional 10x NEBuffer 3.1 was added to adjust for the addition of SDS and Triton X-100 to restore the concentration to 1X NEBuffer 3.1 before adding 400U DpnII and incubated overnight at 37 °C with gentle rocking to digest chromatin. The next morning, DpnII was inactivated by incubation at 65 °C for 20 minutes. A quality check for the completeness of digestion was carried out using 10 µL aliquots taken prior to the addition of DpnII on the previous day and following DpnII heat inactivation. The quality check aliquots were treated with 100ug Proteinase K for 30 minutes at 65 °C and liberated DNA extracted by phenol:chloroform extraction. The DNA was treated with 1ug RNAse A for 15 minutes at 37 °C and then mixed with 6x loading dye and run on a 0.8% Agarose/0.5x TAE gel. If digestion was deemed complete, the remainder of the digested chromatin was prepared for ligation. The ends of the DNA were marked with biotin using a mix of non-biotinylated dCTP, dGTP, dTTP (250 uM final concentration of each) and 250 uM final concentration of biotin-14-dATP and 50U DNA polymerase I Large (Klenow) Fragment (NEB, M0210) within 1x

NEBuffer 3.1, incubated at 23 C for 4 hours in a ThermoMixer (900 rpm mixing; 15 s every 5 mins). The biotinylated DNA was ligated in a reaction mixture containing 1x ligation buffer (NEB, B0202), a final concentration of 1% Triton X-100, a final concentration of 120 ug/mL BSA and 4,000U T4 DNA ligase (NEB, M0202) at 16 C for 4 hours with gentle rocking. Following ligation, the samples were incubated with 500ug Proteinase K overnight at 65 °C. The following day an additional 500ug Proteinase K was added to each tube and incubated at 65 °C for 2 hours to ensure complete digestion of proteins and liberation of ligated DNA. The DNA was isolated by phenol:chloroform extraction using 15 ml PhaseLock tubes and precipitated using 1/10 volume of 3 M sodium acetate, pH 5.2 and 2.5 volumes of cold 100% ethanol. The samples were incubated either on dry ice for 20 minutes or at −80C for 45 minutes-1 hour until the liquid became viscous but not solid. The DNA was pelleted by centrifugation at 15,000 x $g$ for 30 minutes at 4 °C in a chilled fixed-angle rotor. The supernatant was decanted and the pellet allowed to air dry very briefly before resuspending in 1x TLE (10 mM Tris-HCl, pH 8.0, 0.1 mM EDTA, pH 8.0) and concentrated in an Amicon 30 kDa cutoff centrifugal filters (Millipore, UFC503024). The column was centrifuged at 14,000 x $g$ in a tabletop centrifuge for 5 minutes at room temperature, then the column washed at least twice by the addition of 1x TLE and centrifugation again as before except at the last wash, when spun for 10 minutes. Following the last wash, the volume remaining in the column was adjusted to 100ul with 1x TLE and the column inverted in a new collection tube and spun at 1000 x $g$ for 2 minutes to collect the concentrated DNA. The recovered DNA was treated with 1ug RNAse A for 30 minutes at 37 °C and the DNA quantified by Qubit (Thermo Scientific, Q32850). An aliquot of 800 ng of samples was set aside for quality control assay to check completeness of ligation. The quality control assay is a PCR followed by digestion with MboI, ClaI or a double digest as described in the protocol by the Dekker lab[81] and specific for DpnII digested chromatin. With extent of successful ligation confirmed, the biotin was removed from unligated ends in the reaction by incubation of 5ug of the ligated DNA with 25uM each of dATP and dGTP (non-biotinylated) and 15 Units of T4 DNA polymerase in 1x NEBuffer 2.1 and incubated in a thermal cycler set to 20 °C for 4 hours. The enzyme was inactivated at 75 C for 20 minutes and the sample cooled down to 4 °C. Multiple biotin removal reactions (up to 8) were set up and pooled afterwards and the volume adjusted to 500 µL with molecular biology-grade water. The DNA was cleaned and concentrated in an Amicon column by centrifugation at 14000 x $g$ for 5 minutes and washed twice with 400 µL molecular biology-grade water and centrifugation at 14000 x $g$ for 5 minutes. The DNA was recovered by inverting the column into a clean collection tube and centrifugation at 1000 x $g$ for 2 minutes. The volume was adjusted to 132 µL after recovery using molecular biology-grade water and 2 µL removed for sonication quality control as a pre-sonication check. The remaining 130 µL of recovered DNA was transferred to a nonstick 1.5 ml tube (Ambion, AM12475) and sonicated in a QSonica instrument (model Q-800R1-110) at 2 C, amplitude 20% with 10 second pulses and 10 seconds of rest for 6 minutes at 10 W per sample. Following sonication 2 µL was removed as the post-sonication quality control sample. The results of sonication were checked using an Agilent Genomic DNA screentape. If the sheared DNA fell predominantly within the expected range of 200-400 bp the samples could be further size selected using a two-step purification with Ampure XP beads to recover DNA fragments between 100 to 400 bp. The DNA was enriched for fragments containing a ligation junction by capturing biotinylated fragments on MyOne Streptavidin C1 magnetic beads (Thermo Scientific, 65001). The DNA, still captured on the streptavidin beads, was used in the NEBNext UltraII End Prep reaction with the End Prep Enzyme Mix (NEB, E6050L) and incubated for 30 minutes at 20 °C followed by 30 minutes at 65 °C. Following the End Prep reaction, appropriate NEBNext Adaptors for Illumina (NEB, E7350) at 1:15 dilution were ligated using the NEBNext Ultra II Ligation Master Mix and incubated at 20 C for 15 minutes in a thermal cycler without a heated lid to maintain proper temperature. The USER Enzyme was added to the ligation reaction and incubated at 37 °C for 15 minutes in a thermal cycler with the lid temperature at 50 C. The beads were immobilized against a magnetic bar (Dynamag-2, Life Technologies, model # 12321D) and the beads washed twice with Tween wash buffer according to the MyOne Streptavidin C1 protocol, resuspended in 1x BB, made according to the MyOne Streptavidin C1 protocol, and washed twice with 1x TLE, then resuspended in 20 µL TLE. The number of cycles required for the indexing reaction to amplify indexed, biotin-free DNA without introducing PCR bias was determined by setting up a small reaction using 3 µL of beads with bound Adaptor Ligated DNA fragments in a 25 µL indexing PCR reaction using the NEBNext Ultra II Q5 Master Mix, an Index Primer (i7 Primer), Universal PCR primer and 0.225 µL 100x SybrGreen Dye (Invitrogen, S7563). The test PCR reaction was run in a (type of machine, brand, part number). The number of cycles to use in the indexing reaction for the remaining DNA-bound beads was determined by the number of cycles that gives 1/3 the maximum fluorescence. After determining the number of cycles, five 25 µL indexing reactions were set up for each sample and the PCR ran in a BioRad T100 Thermal Cycler with the number of cycles adjusted. The completed indexing PCR reactions for a sample were pooled together in a 1.5 ml non-stick tube and the streptavidin beads immobilized on the Dynamag-2 magnetic bar. The supernatant containing the amplified indexed DNA was transferred to a new 1.5 ml non-stick tube. A 3 µL aliquot of the amplified DNA was saved as a quality control sample for the removal of primer dimers. The rest of the amplified DNA was purified with Ampure XP beads using 1:1.5 ratio of sample to beads and eluted in 35 µL TLE. The pre-purified quality control sample was compared to the post-purified sample using the Agilent Bioanalyzer D1000 Screentape. The molarity of the indexed library was calculated based on the NEBNext qPCR library quantification kit. Libraries were diluted to 4 nM, denatured and prepared for sequencing on the NextSeq 500 or 550 with settings for single-index, paired-end sequencing with 36-42 cycles per end. Data was initially processed using HiC-Pro v.2.10.0[82] visualization and prediction of TADs and loops were done using HiCExplorer v.3.7[83].

## RNA extraction

Extraction of RNA from flash frozen cell pellets or primary tissue was carried out with the miRNeasy kit (Qiagen, 217004). The work surface, pipettes and centrifuge rotors were treated with RNAse Away (Life Technologies, 10328011) prior to beginning. Aliquots of reconstituted DNaseI were prepared from the RNase-free DNase Set (Qiagen, 79254) ahead of time and the appropriate amount of DNaseI in RDD buffer was prepared just prior to starting the extraction. Pellets or tissue were placed on ice and allowed to warm slightly, but not completely thaw. An average pellet or piece of primary embryonic tissue required 700 µL Qiazol. Samples were homogenized in 700 µL QIAzol by pipetting, brief vortexing and applied to Qiashredder columns (Qiagen, 79654). Homogenates were processed immediately after being allowed to incubate at room temperature for 5 minutes. The extraction proceeded with the addition of 140 µL chloroform to the homogenate, which was then shaken vigorously for 15 seconds and allowed to rest at room temperature for 2-3 minutes. The homogenates were then centrifuged for 15 minutes at 12,000 x $g$ at 4 °C. The aqueous phase was transferred to a fresh 1.5 ml tube (typically able to recover 300-350 µL of aqueous phase). To the transferred aqueous phase 1.5 volumes of 100% ethanol was added and mixed by pipetting, then immediately passed through the RNeasy spin column and processed according to the manufacturer's instructions with on-column DNase treatment and the addition of a second wash with Buffer RPE.

## RNA-seq library preparation

Total RNA quality was assessed using the Agilent Tapestation using Agilent RNA analysis screentapes (Agilent Genomics, 5067-5576). RNA with RNA Integrity Number (RIN) scores preferably > 8.0 were used in the preparation of RNA-seq libraries. At minimum, 200 ng of total RNA was used in the reactions for the Illumina TruSeq stranded RNA-seq library preparation kit (Illumina, RS-122-2101) according to the manufacturer's instructions with the modification to use Superscript III Reverse Transcriptase enzyme (Invitrogen, 18080044) during the first strand cDNA synthesis step. Completed libraries were checked for quality and average fragment size using D1000 screen tapes and molar concentration determined using NEBNext qPCR library quantification kit. Libraries were diluted to 4 nM, pooled and denatured according to the instructions for Illumina NextSeq 550/500. Libraries were sequenced on the NextSeq 500 or 550 with settings for single-index, paired-end sequencing with 75 cycles per end. To analyze the data, first adapter contamination was trimmed from the reads using Trimmomatic v.0.36[84] and aligned to the appropriate genome assembly using the STAR aligner v.2.7.1a[85]. Alignment to hg19 used gencode.v19.annotation.gtf and alignment to mm10 used gencode.vM25.annotation.gtf. Differential gene analysis was performed in R v.3.6.3 using DESeq2 v.1.26.0[86]. Resulting Wald test p-values were adjusted using Benjamini-Hochberg approach for all analyses. Surrogate variable analysis was performed with the R package sva v.3.34.0[87].

## Selection of hg19 4C viewpoint primers

Primers for 4C in HoxA region based on locations orthologous to mouse (mm9) HoxA 4 C viewpoints. Primers were designed against NlaIII and DpnII cut sites that produce a fragment near or overlapping the element of interest. The primers were chosen from the 4C-seq primer database from 4cseq_pipe (https://www.wisdom.weizmann.ac.il/~atanay/4cseq_pipe/4c_primer_db_manual.pdf[50]; and are listed in Supplemental Table 12.

## 4C library preparation

Crosslinked cell pellets or crosslinked embryonic tissue was resuspended in 1x Cell lysis buffer (50 mM Tris-HCl, pH 8.0, 140 mM NaCl, 1 mM EDTA, pH 8.0, 10% glycerol, 0.5% IGEPAL CA-630, 0.25% Triton X-100) and nuclei released in a prechilled dounce homogenizer by 10 strokes with a loose glass pestle, 20 minutes rest on ice followed by 40 strokes with a tight glass pestle (Kimble Kontes 885300-0002). The nuclei were pelleted by centrifugation at 2500 x g at 4 °C for 5 minutes, washed twice with 1x NEBuffer 3.1 and pelleted again as before. The nuclei were resuspended in 1x NEBuffer 3.1 and permeabilized by the addition of SDS, incubated for 10 minutes at 65 °C with shaking at 800 rpm on a ThermoMixer C (Eppendorf). The SDS was neutralized with Triton X-100 and incubated at 37 °C for 10 minutes with gentle rocking. Additional 10x NEBuffer 3.1 was added to adjust for the addition of SDS and Triton X-100 to restore the concentration to 1X NEBuffer 3.1 before adding 200 units of NlaIII (NEB, R0125) and incubating overnight at 37 °C with gentle rocking. The next morning, an additional 50 units of NlaIII was added and returned to incubate at 37 °C for 2 hours. Quality control samples of small volume (5ul) were taken before the first addition of NlaIII (undigested) and after the second incubation with NlaIII (digested). These quality control samples were incubated with 1ug RNAse A for 20 minutes at 37 °C followed by incubation with 20ug Proteinase K at 65 °C for 1 hour and heated to 95 °C for 3 minutes to reverse the crosslinks. The samples were then run on a 1% agarose/0.5x TAE gel to visualize the completeness of digestion. If satisfactory, the main reaction tubes were heated to 65 °C for 20 minutes to inactivate the restriction enzyme, take another 5 µL digested control sample, and divided among three prechilled 50 ml conical tubes containing ligation mix (final concentrations of each: 745 µL 10x T4 ligase buffer, 745 µL 10% Triton X-100, 8 µL 100 mg/ml BSA, 5.5 ml dH2O, 2000 units of T4 DNA ligase. The ligation reactions

were incubated overnight at 15 C. The next day, remove 10 µL from each ligation and combine. To the digested control sample and the ligation control sample incubate with RNAse A at 37 °C for 20 minutes and Proteinase K at 65 °C for 1 hour, then heated to 95 °C for 3 minutes and run on a 1% agarose/0.5x TAE gel to assess successful ligation. The crosslinks in the main reactions were reversed by incubating with 300ug Proteinase K at 65 °C overnight. The next day, RNA and protein was removed from the reactions by incubation with 10ug RNAse A at 37 °C for 1 hour and 300ug proteinase K at 65 °C for 2 hours. The digested chromatin was extracted from the ligation reactions by adding an equal volume of premixed phenol:chloroform:isoamyl alcohol (25:24:1) (Ambion, AM9732) and centrifugation at 5000 x g for 10 minutes at 4 °C. A second extraction of the aqueous fraction was performed in a fresh 50 ml conical tube using a volume of chloroform equal to the aqueous fraction and centrifuged at 5000 x g for 10 minutes at 4 °C. The resulting aqueous phase was transferred to a new 50 ml conical tube and the chromatin precipitated by addition of 1/10 volume 3 M Sodium Acetate (pH 5.2) and 2.5 volumes cold ethanol and incubating at −80 C for 16–64 hours. The precipitated chromatin was pelleted by centrifugation at 10,000 x g for 45 min at 4 °C and washed once with cold 70% ethanol and centrifuged again, then air dried after removing the 70% ethanol and each dried pellet separately resuspended in 10 mM Tris-HCl pH 8.0. At this point, 5 µL of the ligated chromatin was saved for quality control and the concentration measured with nanodrop or Qubit (Thermo Fisher, USA). Each ligation reaction was added to the second restriction digest, containing DpnII buffer to 1x final concentration and 150 units DpnII (NEB, R0543). The digestion was incubated overnight at 37 °C with gentle rocking. The following day another 5 µL of the digested chromatin was taken to check the quality of the second digest and run on a 1% Agarose/0.5x TAE gel alongside the ligated chromatin control. When digestion was determined to be adequate from comparing the controls, the reaction was heated to 65 °C for 25 minutes to inactivate the DpnII. Each digestion reaction was transferred to a fresh 50 ml conical tube with 1x T4 Ligase Buffer, water and 3 µL NEB T4 ligase, mixed gently and incubated at 15 °C for 4 hours. Then 1/10 volume 3 M Sodium Acetate pH 5.2 and 1/1000 volume glycogen was added to facilitate DNA precipitation. Cold ethanol at 2.5 volumes was added and the reactions incubated at −80 °C overnight. The next day the reactions were allowed to warm at room temperature for 30 minutes and centrifuged at 10,000 x g for 45 minutes at 4 °C to pellet the DNA. The supernatant was removed and the pellet washed with cold 70% ethanol and centrifuged again as before. The wash was removed and the pellet allowed to air dry. Each of the three reaction pellets were allowed to resuspend in TE for 1 hour at room temperature then pooled into one tube and purified across three Qiaquick PCR purification columns (Qiagen, 28104), eluted in the provided Elution Buffer and combined. The concentration was determined by nanodrop or Qubit. The quality of the second ligation was assessed by running ~500 ng of the purified product on a 1% agarose/0.5x TAE gel. Amplification of specific viewpoints was done by inverse PCR with primers designed against NlaIII and DpnII cut sites that produce a fragment near or overlapping the element of interest. The amplification reaction consisted of 50 ng template, Roche Long Template Buffer 1 (Roche Applied Bioscience, Sigma Aldrich, 11681834001), 200 nM dNTPs (Roche Applied Bioscience, Sigma Aldrich 4829042001), 200 nM each of forward and reverse primer for the specific viewpoint and 0.7 µL Long Template Polymerase Mix. To reduce amplification bias, eight to 16 replicate PCR reactions were performed per viewpoint. The replicate reactions were pooled and used in the subsequent indexing PCR reaction. Pooled replicate viewpoint amplification reactions were purified using QIAquick columns according to the manufacturer's instructions, quantified by nanodrop or Qubit and five to eight indexing PCR reactions were set up per viewpoint. The indexing PCR reaction used 50–100 ng template, Roche Long Template Buffer 1, 200 nM dNTPs, 200 nM i5

and i7 index primers and 0.7 μL Long Template Polymerase Mix. The indexing primers carrying sequences recognized as [truseq 701-…, 501-…] are listed in Supplemental Table. The replicate indexing reactions were pooled and QiaQuick columns followed by Axygen bead cleanup with 1:1 ratio AxyPrep Mag PCR Clean Up beads (Axygen, 14223151). The indexed libraries, now in their final form, were checked for quality using the Agilent Tapestation (Agilent, USA) with D1000 Screen Tape. The molar library concentration was assessed using NEBNext Library Quant Kit for Illumina (NEB, E7630). Libraries were diluted to 4 nM, pooled and denatured for sequencing on the Illumina NextSeq 500 or 550 as directed and sequenced using the settings for 75 bp single-end, dual index sequencing. Data was processed as described at https://github.com/cotneylab/Mouse-HOXA-4C-Seq. Briefly, the biological samples were first demultiplexed by viewpoint using cutadapt[88], aligned to the genome using bowtie2 and analyzed using a version of r3Cseq[89]) modified to allow visualization of plots over larger distances (modified version available at https://github.com/cotneylab/r3Cseq).

### Microcomputed tomography imaging and analysis of embryos
In preparation for microcomputed tomography (microCT) imaging, fixed E18.5 embryos were briefly rinsed in 70% ethanol and placed individually on custom styrofoam beds that held the embryos in an upright position. Embryos were then individually scanned using a Skyscan model 1275 benchtop microCT (Bruker BioSpin Corporation) using the following parameters: an isotropic resolution of 18 microns, 40 kV, 180 microAmp, 45 ms exposure, 0.3˚ rotation step, 180˚ rotation, and 4 frame averaging. No filter was used. 16-bit raw images from all scans were reconstructed to 8-bit bmp files using NRecon software v1.7.4.2 (Bruker BioSpin Corporation). Reconstructed scan data were then imported into Drishti volume exploration software (version 2.63)[90] for 3D rendering. Separate rendering settings were optimized for visualization and phenotypic assessment of both mineralized and soft tissues.

To visualize the pterygoid/basisphenoid and tympanic regions in isolation, the majority of the craniofacial bone around these structures was masked using clipping planes. Then, the morphological operations (MOP) carve function was used to remove remaining bone from around the complex. To make rotational movies of the complexes to aid their inspection, the Keyframe Editor function of Drishti was employed. For this, a new rotational axis was assigned for each volume and the initial keyframe set to mark the starting view of the rotation. The desired end of the rotation was set using the Bricks Editor function and a second keyframe set. All interpolated keyframes between the starting and ending keyframe were then saved as an image sequence in png format. Image sequences were then opened in Adobe Photoshop 2021 and rendered in mp4 format. Selected images from the renderings were saved and optimized for contrast, color, and background using Adobe Photoshop.

### Copy number detection through shallow whole-genome sequencing
Fetal gDNA was extracted and isolated upon biopsy from fetal tissue on QIAcube, using the QIAamp DNA Blood Mini Kit (Qiagen). Shallow whole genome sequencing for copy number variant analysis was performed on extracted gDNA using the NEXTflex Rapid DNA Sequencing kit (Bioo Scientific). The normalized libraries were sequenced on a HiSeq 3000 platform (Illumina) and data-analysis was performed according to Raman et al. Raman et al., ($year$)[91] with a 400 kb resolution.

### Mendeliome analysis
Extracted fetal gDNA was further used to perform whole exome sequencing (WES). The coding exons and flanking intronic regions were enriched with the SureSelectXT Low Input All Exon v7 kit (Agilent Technologies) followed by dual index, paired-end (2 x 150 bp) sequencing on a NovaSeq 6000 platform (Illumina). Raw sequence reads were processed using an in-house developed pipeline. Reads were aligned to GRCh38/hg38 and data analysis was performed by Center for Medical Genetics at Ghent University Hospital. Variant analysis was limited to 4732 genes related to a known disease in Mendeliome panel version 5. At least 90% of the coding regions of the included genes had a minimum coverage of 20x. A trio-analysis was performed. Only variants classified as class 4 (possibly pathogenic) and class 5 (pathogenic), if clinically relevant, were reported.

### Targeted long-read sequencing
DNA for targeted sequencing was isolated from blood using the Monarch HMW DNA Extraction Kit for Cells & Blood (NEB) following the manufacturer's protocol. Approximately 1.5 μg of genomic DNA was sheared using a Covaris g-TUBE to approximately 10 kbp by centrifuging at 4400 x $g$ and inverting twice. Libraries for sequencing were prepared using the ONT SQK-LSK110 kit with the following modifications: the DNA repair step was held at 60 °C for 30 minutes instead of 5 minutes, and the ligation step was performed for 1 hour at room temperature instead of 10 minutes. Adaptive sampling[92] was configured to target the duplicated region on chromosome 7 (chr7:24,500,000-29,000,000), a second duplication identified by clinical testing on chromosome 11 (chr11:123,500,000-123,700,000) and three control regions: COL1A1 (chr17:50150000-50250000), FMR1 (chrX:147850000-148000000), and AR (chrX:67500000-67800000) (all targets in hg38 coordinates). Sequencing was allowed to run for 72 hours and FASTQ files were generated using Guppy 5.0.11 (Oxford Nanopore) using the high accuracy model then aligned to GRCh38 using minimap2[93]. Reads were visualized using IGV (Robinson et al. 2022).

### iPSC generation and neural crest induction
Patient and control iPSC lines used in this study were generated by the Cincinnati Children's Pluripotent Stem Cell Facility. iPSC colonies were maintained in mTesR medium (StemCell Technologies) and cultured on Matrigel (Corning, 354277). Cells were typically passaged once a week using GCDR (StemCell Technologies). iPSCs were differentiated into NCCs with the STEMDiff Neural Crest Differentiation Kit (StemCell Technologies) according to the company instructions. Briefly, one well of iPSCs was detached with Accutase (ThermoFisher) into single cells. Cells were resuspended in provided medium containing 10 μM Y-27632 (Cell Signaling, 13624S) and plated at $8.6 \times 10^4$ cells/cm² on Matrigel-coated 12-well plates. Cells were cultured with daily medium changes without Y-27632. On day 6, cells were passaged with Accutase into single cells (NCC P1).

## Data availability
The data generated from this study including HiC, 4C-Seq, and RNA-Seq in both mouse and human samples have been deposited in the Gene Expression Omnibus under accession code GSE198297. The raw fastqs from CS17 HiC are available under restricted access due to the genome-wide coverage of sequence from this sample are available at dbGAP under accession code phs002008. Signal tracks for these experiments are available through track hub functionality at the UCSC Genome Browser (https://cotneylab.cam.uchc.edu/~jcotney/CRANIOFACIAL_HUB/Craniofacial_Data_Hub.txt), as a public session (https://genome.ucsc.edu/cgi-bin/hgTracks?hgS_doOtherUser=submit&hgS_otherUserName=Jcotney&hgS_otherUserSessionName=Human%20HOXA%20LCR) or at the Cotney Lab website (https://cotney.research.uchc.edu/hoxa_gcr/). Raw data generated in this study are used in Fig. 3, Fig. 4, Fig. 5, Fig. 6, and Fig. 7. The following datasets were downloaded from the Gene Expression Omnibus (GEO): Series GSE197513: "Transcriptomic Atlas of Embryonic Human Craniofacial Development" (Yankee et al. 2023); Series GSE105028: "Architectural proteins and pluripotency factors cooperate to orchestrate the

transcriptional response of hESCs to temperature stress"[94]; Series GSE145327: "CTCF ChIP-seq in undifferentiated H9 hESCs and H9-derived CNCCs"[46]; and Series GSE104173: "Expression data from retinoic acid-induced differentiation of human embryonic stem cells (hESCs)". Plots generated from HiC data using the HiC Browser hosted at Northwestern University (3dgenome.fsm.northwestern.edu/view.php) included Adrenal Gland, Cortex, Right Ventricle[95] GM12878, IMR90[45] and Liver[96].

## Code availability

Code used to analyze 4C-seq data is available at github.com/cotney/Mouse-HOXA-4C-Seq. The most recent HiC pipeline is at github.com/awilderman/HiC. Code used in the analysis of human and mouse superenhancers is available at github.com/awilderman/Thesis.

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

## Acknowledgements

We would like to thank the laboratory of James P. Noonan for providing images of the HACNS50 enhancer reporter experiments. We would also like to thank the staff of the University of Connecticut Computational Biology Core and the High Performance Computing Center for assistance with package installation and software/hardware support. This work was supported by the Cincinnati Children's Hospital Medical Center Pluripotent Stem Cell Facility. This work was funded by the National Institutes of Health DP DP5OD033357 (DEM), NIDCR R35 DE027557 (SAB), NIDCR 1R01DE028945 (JC), NIDCR 1R03DE028588 (JC), and NIGMS 5R35GM119465 (JC). TC is supported by a Stowers Family Endowment. This study makes use of data generated by the DECIPHER community. A full list of centers who contributed to the generation of the data is available from https://deciphergenomics.org/about/stats and via email from contact@deciphergenomics.org. Funding for the DECIPHER project was provided by Wellcome.

## Author contributions

Conceptualization A.W. and J.C. Investigation A.W., E.D., M.B., T.N.Y., E.W.W., D.E.M., M.G., K.M.B., and T.C. Formal Analysis A.W., E.D., M.B., E.W.W., K.M.B., S.A.B., T.C., S.G., S.V., and J.C. Resources N.G., E.R., J.V.D., S.J., R.W.S.. Writing - Original Draft A.W. and J.C. Writing––Review and Editing A.W., E.D., M.B., T.N.Y., E.W.W., E.R., J.V.D., S.J, D.E.M., M.G., R.W.S., S.V., K.N.W., S.A.B., T.C. and J.C. Supervision S.V., S.A.B., T.C. and J.C. Funding Acquisition D.E.M., S.A.B., T.C., and J.C.

## Competing interests

DEM holds stock options in MyOme. DEM is on a scientific advisory board at ONT, is engaged in a research agreement with ONT, and they have paid for him to travel to speak on their behalf.

## Additional information

[1]Graduate Program UConn Health, Farmington, CT, USA. [2]Center for Medical Genetics, Department of Biomolecular Medicine, Ghent University, Ghent, Belgium. [3]University of Connecticut School of Dental Medicine, Farmington, CT, USA. [4]Department of Genetics and Genome Sciences, University of Connecticut School of Medicine, Farmington, CT, USA. [5]Department of Obstetrics, Women's Clinic, Ghent University Hospital, Ghent, Belgium. [6]Department of Pathology, Ghent University, Ghent University Hospital, Ghent, Belgium. [7]Department of Pediatrics, Division of Genetic Medicine, University of Washington, Washington, WA, USA. [8]Seattle Children's Hospital, Seattle, WA 98195, USA. [9]Department of Laboratory Medicine and Pathology, University of Washington, Seattle, WA 98195, USA. [10]Brotman Baty Institute of Precision Medicine, University of Washington, Seattle, WA 98195, USA. [11]Division of Developmental Biology, Cincinnati Children's Hospital Medical Center, Cincinnati, OH, USA. [12]Department of Pediatrics, University of Cincinnati College of Medicine, Cincinnati, OH, USA. [13]Steve and Cindy Rasmussen Institute for Genomic Medicine, Nationwide Children's Hospital, Columbus, OH, USA. [14]The Abigail Wexner Research Institute, Nationwide Children's Hospital, Columbus, OH, USA. [15]Department of Pediatrics, The Ohio State University School of Medicine, Columbus, OH, USA. [16]Division of Human Genetics, Cincinnati Children's Hospital Medical Center, Cincinnati, OH, USA. [17]Department of Oral & Craniofacial Sciences, University of Missouri Kansas City, Kansas City, MO, USA. [18]Department of Pediatrics, University of Missouri Kansas City, Kansas City, MO, USA. [19]Institute for Systems Genomics, University of Connecticut, Storrs, CT, USA. ✉e-mail: cotney@uchc.edu

