## [Transparent Peer Review File · Nature Communications]

A distant global control region is essential for normal expression of anterior HOXA genes during mouse and human craniofacial development.Reviewer #1 (Remarks to the Author: Overall significance):

Wilderman et al. extended their previous study of global epigenomic mapping of DNA regulatory elements in different craniofacial tissues/cell types to this work in which they identified a local, long-range enhancer region that drives HOXA gene cluster expression in both human and mouse embryonic craniofacial tissues. They found that homozygous deletion of this region in mice led to skull defects, orofacial clefts and perinatal lethality. They were able to find one example of human fetus in which a partial overlapping deletion also caused severe craniofacial abnormality and preterm pregnancy termination. The reviewer finds the work interesting as to identify a long-distance non-coding regulatory region potentially linked to human developmental defect, which has a potential broad readership in genome organization, transcription regulation, and developmental biology et al.

Reviewer #1 (Remarks to the Author: Strength of the claims):

However, several conclusions are not well supported by existing evidence and therefore need further experiments to substantiate. My primary concerns are:

1. Although they identified a mouse syntenic region responsible for Hoxa cluster expression, they did not provide convincing evidence about the human counterpart as bona fide tissue specific enhancers. The region is ~600kb and contains several enhancer-like elements (Fig.2b-c). The authors cited several studies and claimed that deleting a single enhancer element might cause mild gene expression change. But in some genes (Sox2 or Klf4), a single enhancer element could control majority of transcription activity (Li et al, 2014; Zhou et al, 2014; Xie et al, 2017). Given that the individual enhancer element within this large region (~600kb) has non-identical spatial expression pattern (Fig.2c), the functional role of these elements is unclear at the moment, constituting one major weakness of this study. Similarly, in the single human fetus example, the authors claimed a partial overlapping between the pathogenic deletion with the “superenhancer” they discovered. Failing to identify and pinpoint the constitute enhancer element driving the pathological phenotype leaves much uncertainty to the conclusions they draw.
2. The CNCC cells used in this study should be a powerful model for functional and mechanistic studies of the enhancer elements. However, the authors claimed that “they were unable to remove both alleles” likely due to “harsh manipulation” or “essential for human stem cells”. Given the tissue-specific nature of the regulatory element and the late phenotype of homozygous deletion in mice, the failure to obtain the homozygous deletion could be due to technical issues. The authors should take the effort to generate homozygous deletion clones from hPSCs and differentiate them into CNCC cells to verify and corroborate their findings in mouse or consider using orthogonal strategies like CRISPRi or CRISPRoff to revisit the function of individual element or as a whole.
3. The authors claimed discovering a craniofacial specific enhancer region. In Fig.7 Supplement 1 the locus deletion also reduced HoxA cluster expression in limb and heart. Is it because the locus is not that ‘specific’ or the deletion is too large to mask the tissue specificity? More attention should be given to their claims.
4. This study relies on Hi-C to examine the contact between the enhancer and HoxA gene. The observation of the inversion CNCC cell line has even higher contact frequencies is intriguing, given the CTCF site is also inverted. The reviewer wonders what this means in terms of the framework of loop extrusion model and also the chromatin contact in the 3D physical space? Is the distance smaller or the frequency higher? This relates to a fundamental question of understanding ‘contact frequency’ in Hi-C and its functional role in long-range gene activation. Although detailed characterization may go beyond the scope of the current study, the authors should consider performing 3D DNA FISH to quantify the distance distribution of enhancer-gene in WT and

inverted (or deleted) conditions. An illustration model to explain the putative chromatin extrusion/looping formation could further increase the readability of this manuscript to readers not only in developmental biology but also in genome organization and transcription regulation.

Other smaller concerns or minor points:

1. In abstract, missing punctuation before “Mice lacking this ...”.
2. In Fig.6A legend, please specify the mouse genome build mm10 or mm39?
3. In line 297, they claim that “we did not notice major difference in morphology or gene expression”. But in the vein diagram in Fig.4 Supplement 3, the differentially expressed genes approach ~25% or more. Having >1000 genes changing their expression is certainly not ‘minor’. Do I miss something to their phrasing?
4. In line 395, the authors claim that their deletion mice have phenotypes essentially identical to *Hoxa2* null mice. Their study also showed much wider reduction of multiple Hox gene expression in the Hoxa spectrum. The authors may not overclaim their phenotype.
5. This study mainly focused on deleting the enhancer elements in cis but did not go further identifying the key transcription factors binding to these enhancers in trans. Although pinpointing the exact role of TF ensemble on these enhancers require additional work (deletion TF binding site or knocking down TFs), the authors should at least bioinformatically analyze the putative TFs on these constitute enhancer elements within the 600kb region.

Reviewer #2 (Remarks to the Author: Overall significance):

The manuscript titled “A distant global control region is essential for normal expression of anterior HOXA genes during mouse and human craniofacial development” by Wilderman et al. identified a super enhancer region located 1.5Mb upstream of HoxA cluster controlling the expression of HoxA genes in craniofacial development. The regulation of Hox genes during development is interesting and the findings presented in this manuscript are novel. However, there are three major issues that need to be addresses before the manuscript could be considered for publication.

Reviewer #2 (Remarks to the Author: Impact):

The findings in the manuscript is interesting but more experiments/analysis need to be done to support the claims of the authors.

Reviewer #2 (Remarks to the Author: Strength of the claims):

1. The rationale of why the authors focus on this one super enhancer regions out of are not well explained. They select this one large enhancer clusters based on it “...containing the greatest number of confirmed craniofacial enhancers...”. What about other regions? How many enhancers are in them? Are there much less enhancers in the other 36 regions compared with this one selected region? Without addressing these questions, it feels like the authors cherry picked a super enhancer cluster to study and raises the question on the validity of the selection process.

2. The enhancer cluster the authors focused on is fairly large (600Kb). The inversion experiment in hESCs and the deletion experiment in mice are focusing on this 600Kb region rather than specific enhancers inside. Mutating such a large region has the potential risk of generating or masking phenotypes specific associated with the true enhancers. The authors should generate deletion mutants of individual enhancers or compound enhancer deletion mutants in hESCs by only mutating the previously characterized enhancers. Fine mapping and individually deleting the

enhancers in this region could provide insights in whether the enhancers are truly functional and whether they have additive or synergistic roles in regulating HoxA genes.

3. Does deleting individual enhancers or even the whole 600Kb region change genome architecture near Hox genes? The authors should perform Hi-C or 4C-seq in the deletion mutant and potentially in the individual enhancer mutants.

Reviewer #3 (Remarks to the Author: Overall significance):

In this paper, the authors identify super enhancer elements across the human genome using their unique previously published datasets of epigenomic profiling from human embryonic craniofacial tissue. One super enhancer (SE) element was prioritised for follow-up due to the presence of many previously characterised facial enhancers from both the human and mouse genome. This SE region interacts with the HOXA gene cluster in in vitro derived cranial neural crest cells and in vivo. Deletion of a larger region encompassing the SE region causes craniofacial dysmorphology when mutated in mice, phenocopying loss of the *Hoxa2* gene. Loss of the SE region impacts anterior Hox genes specifically in the developing face consistent with the tissue-specificity of this regulatory region. A large mutation that encompasses the 3' end of the SE region is also associated with craniofacial abnormalities in a human fetus. Together this paper reveals a non-coding region important for Hoxa gene expression and normal craniofacial development. The results will be of interest to many in the field as this locus represents another example of extreme-long range regulation in development, and implicates a large cluster of enhancers / super enhancer.

There are some caveats to the study. In their mouse models, the authors generated a very large deletion that encompasses the SE and surrounding sequence including TAD boundaries. Therefore, it remains to be confirmed whether loss of the enhancer cluster or other adjacent sequences is responsible for the impact on HOXA gene expression they observe. The SE cluster is identified by the ROSE algorithm based on K27ac data, but the raw ChIP-seq profiles are not shown for this region, and individual enhancers within the SE cluster are not identified. These regions were not tested for activity in in vitro derived CNCCs, and enhancer function was neither determined by deletion due to potential lethality of homozygous SE ablation in hESCs. Several lines of evidence converge on the importance of the 3' end of the SE for looping to the Hoxa locus, and ablation is associated with human craniofacial malformation. A more careful dissection of the putative regulatory elements at the 3' end of the SE would aid interpretation of these results.

Reviewer #3 (Remarks to the Author: Strength of the claims):

Major and minor comments are outlined below.

Major comments

1. Figure 2B. Within the SE region, can the authors identify individual enhancers from the epigenomic profiling data? Are some of these active enhancers active by luciferase assay in human CNCCs?

2. The CNVs described in Figure 2 Supp1 are extremely large, impacting many genes including the HOXA gene cluster. Caution should be taken here, and the DECIPHER data doesn't support the statement "these findings strongly suggest that this superenhancer region is specific for craniofacial development and associated with craniofacial abnormalities" as the patient phenotypes are variable (with or without craniofacial abnormalities) and the CNVs impact many genes in addition to the SE region. This is discussed later in the manuscript, but is perhaps misleading in the earlier discussion of this data.

3. The logic for deleting the entire TAD in Figure 4 is somewhat unclear. By designing the guides either side of a predicted TAD boundary, the locus architecture may in fact be more greatly perturbed. More targeted deletion of the super enhancer (or elements within) would make interpretation of the results less ambiguous. Could the authors discuss further why they believe they are able to delete the enhancer cluster in mouse but not human? Are Hoxa mutations tolerated in hESCs?
4. What was the impact of heterozygous SE deletion in human CNCCs on expression of the HOXA genes? Are there SNPs within the HOXA genes that allow you to look at allele-specific expression from the wildtype versus SE-mutant alleles by qPCR or ddPCR?
5. "All clones we obtained grew relatively normally in ESC culture conditions and we did not notice any major difference in morphology or gene expression". The gene expression analysis appears rudimentary in Figure 4 Supplement 3-4 – more sophisticated differential gene expression analysis and presentation, e.g. using DEseq and plotting base-mean expression versus foldchange with indication of significance would be more illustrative of the change of magnitude of expression (for example similar to the plot the authors have shown in Fig7A). Which classes of genes are changing their expression, if any?
6. "Given these results we reasoned that HOXA gene expression might be maintained at normal levels in these cell lines through upregulation of the cluster that remained in cis with the superenhancer region". This statement is ambiguous – are the authors suggesting that the SE activity increases with loss of one copy of the region? This could be tested by H3K27ac ChIP, and by allele-specific PCR strategies mentioned above. It would be interesting to determine if inversion of the locus leads to an upregulation of the HOXA genes compared to the heterozygous lines given that you observe increased long-range interactions across the domain for the INV line.
7. "the robust expression of HOXA despite these changes precluded us from making any
8. determination of the role of this region in craniofacial related biology". The authors haven't illustrated that HOXA gene expression isn't significantly changed in the mutant CNCCs, a plot for this (RNA-seq or qPCR) would be informative.
9. The phenotypes of GCR deletion are not fully penetrant, does this correlate with a variable impact on Hoxa gene expression earlier in development between embryos? Are the Hoxa2^{-/-} line phenotypes fully penetrant?
10. Figure 7A. What percentage of Hoxa2, Hoxa3 and Hoxaas2 expression is lost in the GCR mutant embryos? No other gene expression changes are detected indicating that this effect is direct. It may be interesting to look at a later stage of development to determine the impact on expression of Hoxa target genes.
11. Figure 7 Supplement 1. Several Hoxa genes appear to be impacted in the GCR homozygous deletion lines in limb tissue. Are there other limb enhancers that may be deleted in the large GCR deletion?
12. For Figure 7 Supplement 2. Could the authors provide images of the HiC data rather than schematics.
13. Figure 8 Supplement 2. How do the 550kb deletion coordinates compare to the GCR deletion and to the SE you describe?

Minor comments

14. It would be informative to see the intersection of the SEs detected across the 17 human craniofacial samples as an Upset plot or Venn diagram to know how stage-specific the elements are that have been detected.
15. An introduction to the ChromHMM states would aid the reader starting from Figure 2, including perhaps a key.
16. Several of the enhancers in Figure 2C appear to be from mouse, it would be useful to have an indication of the conservation of these elements to the human sequence. Are all of the tested enhancers overlapping with predicted enhancer elements from your epigenomic profiling in human craniofacial tissues? It may be useful to see the raw ChIP-seq traces for these enhancers.
17. It would be useful to have the SE region also marked on Figure 2 Supplement 1b along with the enhancer regions of interest within the SE. Which of these enhancers overlaps with the gnomAD variant-depleted region?
18. The text mentions 10 patients with a single CNV at the HOXA locus which may be causative – could these be indicated in Figure 2 Supp1 – how many of these are the patients with a craniofacial phenotype?
19. A comparison of the craniofacial tissues in which the tested enhancers from the HOXA SE are active (Figure 2C) to phenotypes detected in patients from DECIPHER could be informative.
20. Figure 2 Supp 3 is very hard to interpret as the boxplots are so thin. Perhaps a separate plot per gene would be easier to read.
21. Figure 3. It would be useful to indicate the regions that interact more strongly in CNCC versus ESC. Perhaps by subtraction maps.
22. Figure 4B. “these closely spaced CTCF binding sites directly coincided with the boundary between strongly active and strongly repressed chromatin signals in both CNCCs and primary craniofacial tissues”. CTCF sites don’t appear to be annotated in the figure.
23. Figure 7B shows rather variable expression for several genes and it is difficult to interpret. Another representation of this data may be easier to interpret, e.g. boxplot or volcano plot with indication of significant changes?

Reviewer #3 (Remarks to the Author: Reproducibility):

In several cases, figures are difficult to interpret, lack sufficient annotation or show schematics as opposed to data. Updating these figures (suggestions outlined above) will greatly improve the manuscript.

Reviewer 1

1. Although they identified a mouse syntenic region responsible for Hoxa cluster expression, they did not provide convincing evidence about the human counterpart as bona fide tissue specific enhancers. The region is ~600kb and contains several enhancer-like elements (Fig.2b-c). The authors cited several studies and claimed that deleting a single enhancer element might cause mild gene expression change. But in some genes (Sox2 or Klf4), a single enhancer element could control majority of transcription activity (Li et al, 2014; Zhou et al, 2014; Xie et al, 2017). Given that the individual enhancer element within this large region (~600kb) has non-identical spatial expression pattern (Fig.2c), **the functional role of these elements is unclear at the moment, constituting one major weakness of this study**. Similarly, in the single human fetus example, the authors claimed a partial overlapping between the pathogenic deletion with the “superenhancer” they discovered. **Failing to identify and pinpoint the constitute enhancer element driving the pathological phenotype leaves much uncertainty to the conclusions they draw**.

We have adjusted the language throughout the manuscript to better reflect the concept of tissue-specific superenhancers as clusters of enhancers appearing to be specifically co-activated and potentially operating as a unit in a specific tissue, but not precluding the activity of individual enhancers in more than one developmental time or tissue.

While it is accurate that there are cases in which a single enhancer element appears to control the majority of transcriptional activity, redundancy and cooperativity among individual enhancer elements within a superenhancer is frequently the mode of regulation. This is reviewed extensively by Kvon et al 2021. In the case of a conserved superenhancer with potential influence on critically important developmental genes, we expect, as others have found, that each individual enhancer element contributes a small effect upon gene expression, and that notable phenotypes may only be distinguished with the deletion of two or more individual enhancer elements from the superenhancer. Within the window we are studying (chr7:25,295,587-25,921,144; hg19) the number of individual enhancer segments contained within the superenhancer loci alone average 119 in human craniofacial tissue (depending on stage and sample, ranging from 58-215). In the H9-derived CNCC model cell system, there are at minimum 72 individual enhancer segments in the superenhancers. Even limiting deletion targets to strong enhancer states from the 25-state model (13_EnhA1, 14_EnhA2, 15_EnhAF) found within the superenhancers results in a minimum of 30 individual strong enhancer elements. To create deletions of pairs of enhancers, then would require thousands of individual clones.

Likewise, within the ~520kb region targeted for deletion in mouse (chr6:50,673,614-51,196,805; mm10) strong enhancer states from the 18-state model (8_EnhG2, 9_EnhA1, 10_EnhA2) found within the superenhancer regions identified at stages E9.5-E15.5 would still yield between 67 and 139 individual strong enhancer elements to target for deletion, which would be prohibitive for creating mouse knockout lines carrying each possible combination of pairs of enhancers. Therefore, the strategy of deleting the entire superenhancer region to gauge any effect on HOXA gene expression was our primary approach.

Furthermore, we have identified now two patients effected by copy number variants at this region much smaller than those described in DECIPHER. The second patient we have characterized iPSCs and shown that indeed HOXA gene expression is perturbed. This is the best possible example that we could think of in existing human patients.

2. The CNCC cells used in this study should be a powerful model for functional and mechanistic studies of the enhancer elements. However, the authors claimed that “they were unable to remove both alleles” likely due to “harsh manipulation” or “essential for human stem cells”. Given the tissue-specific nature of the regulatory element and the late phenotype of homozygous deletion in mice, the failure to obtain the homozygous deletion could be due to technical issues. The authors should take the effort to generate homozygous deletion clones from hPSCs and differentiate them into CNCC cells to verify and corroborate their findings in mouse **or consider using orthogonal strategies like CRISPRi or CRISPRoff to revisit the function of individual element or as a whole**.

We agree that this is indeed a strange result. At this point we have screened several hundred clones without recovering a homozygous deletion. However we have now successfully deleted both alleles of the TAD boundary and see little effect on target gene expression. We have now also further investigated gene expression of the HOXA gene cluster in the del/inv line. Surprisingly we find that there is strong allele specific expression in this line. This indicates the allele lacking the GCR does not robustly express HOXA genes in the CNCC system and that the remaining inverted allele is providing more robust activation of these target genes. This is shown in Figure 4 Supplement 8.

3. The authors claimed discovering a craniofacial specific enhancer region. In Fig.7 Supplement 1 the locus deletion also reduced HoxA cluster expression in limb and heart. **Is it because the locus is not that 'specific' or the deletion is too large to mask the tissue specificity?** More attention should be given to their claims.

Our claim is that this is a craniofacial specific superenhancer. While there are indeed some small segments that are active in other tissues, the larger super-enhancer state is not observed. Furthermore our RNA-Seq analysis does not indicate statistically significant differences in limb or heart and we do not observe any phenotypes in those tissues. We have updated the supplemental figure to reflect the lack of statistical significance. We have also added microCT scans of the limbs to demonstrate no phenotype.

4. This study relies on Hi-C to examine the contact between the enhancer and HoxA gene. The observation of the inversion CNCC cell line has even higher contact frequencies is intriguing, given the CTCF site is also inverted. The reviewer wonders what this means in terms of the framework of loop extrusion model and also the chromatin contact in the 3D physical space? Is the distance smaller or the frequency higher? This relates to a fundamental question of understanding 'contact frequency' in Hi-C and its functional role in long-range gene activation. Although detailed characterization may go beyond the scope of the current study, the authors should consider performing 3D DNA FISH to quantify the distance distribution of enhancer-gene in WT and inverted (or deleted) conditions. **A illustration model to explain the putative chromatin extrusion/looping formation could further increase the readability of this manuscript to readers not only in developmental biology but also in genome organization and transcription regulation.**

The addition of Figure S13 illustrates the change in orientation of CTCF binding sites that may be influencing the strength of interaction between loci in the inversion line. As for 3D DNA FISH to quantify distance distribution between enhancer and promoter, given findings in a recent publication, this may not be as illustrative as thought under previous paradigms of enhancer action. Zuin et al. (doi:10.1038/s41586-022-04570-y) used an elegant system to illustrate the nonlinear relationship between contact probability and expression. They propose a model whereby the number of intermediate regulatory steps required for transcription initiation can account for high rates of transcription despite low contact probabilities. If the intermediate steps are the determinants, I do not know an effective way (within the scope of this work) to find what they are or how many there are required to influence HOXA gene expression.

Other smaller concerns or minor points:

1. In abstract, missing punctuation before "Mice lacking this ...".
Corrected.

2. In Fig.6A legend, please specify the mouse genome build mm10 or mm39?
Specified mm10, thank you for catching that item that escaped copyediting.

3. In line 297, they claim that "we did not notice major difference in morphology or gene expression". But in the vein diagram in Fig.4 Supplement 3, the differentially expressed genes approach ~25% or more. **Having >1000 genes changing their expression is certainly not 'minor'**. Do I miss something to their phrasing?
We have clarified our presentation of the comparison of gene expression between the WT and INV cell lines. The major source of differential expression is the stage of differentiation, WT and INV H9 cell lines being more similar to each other than to their respective differentiated CNCCs. Differential expression between WT and INV CNCCs is only found for 96 genes.

4. In line 395, the authors claim that their deletion mice have phenotypes essentially identical to Hoxa2 null mice. Their study also showed much wider reduction of multiple Hox gene expression in the Hoxa spectrum. **The authors may not overclaim their phenotype.**

While we agree that we see effects on other flanking Hoxa genes not reported in those studies, the fact remains that for all the phenotypes we examined they were strikingly similar to those reported for the Hoxa2 null mice. Our findings have been recently confirmed by others (Kessler et al 2023).

5. This study mainly focused on deleting the enhancer elements in cis but did not go further identifying the key transcription factors binding to these enhancers in trans. Although pinpointing the exact role of TF ensemble on these enhancers require additional work (deletion TF binding site or knocking down TFs), the authors should at least **bioinformatically analyze the putative TFs on these constitute enhancer elements within the 600kb region**.

We have bioinformatically analyzed the transcription factor binding sites enriched within the strong enhancer states of the mouse craniofacial superenhancer as well as embryonic heart and embryonic limb superenhancers identified within the same 600kb region. These overlapping superenhancers are illustrated in Figure 7 Supplement 1. We report the transcription factor binding motifs enriched in the strong enhancer states from each relevant tissue superenhancer in Figure 7 Supplement 2. Despite shared enriched motifs for GRE, PBX1 and Pitx1 between face and limb, the gene expression affecting the HOXA cluster and phenotype are craniofacial-specific and do not appear to affect the limbs.

Reviewer 2

1. The rationale of why the authors focus on this one super enhancer regions out of are not well explained. They select this one large enhancer clusters based on it "...containing the greatest number of confirmed craniofacial enhancers...". What about other regions? How many enhancers are in them? Are there much less enhancers in the other 36 regions compared with this one selected region? Without addressing these questions, it feels like the authors cherry picked a super enhancer cluster to study and raises the question on the validity of the selection process.

We have divided the first section of the results into two, the first addressing the identification of superenhancers in human embryonic craniofacial tissue and the second addressing craniofacial specific superenhancers more specifically. This includes a more detailed walk through the logic and process of looking at craniofacial specific superenhancers within gene deserts. We used a defined size for gene desert as $\geq 500\text{kb}$ without overlapping a protein-coding sequence. Re-analysis using this definition did not alter the results substantially. A table showing the VISTA elements with positive staining in craniofacial tissue within the gene deserts which contain craniofacial specific superenhancers has been added.

2. The enhancer cluster the authors focused on is fairly large (600Kb). The inversion experiment in hESCs and the deletion experiment in mice are focusing on this 600Kb region rather than specific enhancers inside. Mutating such a large region has the potential risk of generating or masking phenotypes specific associated with the true enhancers. **The authors should generate deletion mutants of individual enhancers or compound enhancer deletion mutants in hESCs by only mutating the previously characterized enhancers.** Fine mapping and individually deleting the enhancers in this region could provide insights in whether the enhancers are truly functional and whether they have additive or synergistic roles in regulating HoxA genes.

One of the major questions that we hoped to address with this work was the basic super-enhancer concept. As we detailed in our analysis, and is true generally from other studies, super-enhancers typically contain one or more genes. This makes deleting an entire super-enhancer difficult to study as a gene would also be removed. We have achieved this and demonstrate a strong tissue specific phenotype. We have now deleted only the strong TAD boundary most proximal to the HOXA gene cluster in hESC. This deletion has very little effect on HOXA gene expression in CNCCs. While we recognize we have not shown that a single enhancer may cause this very specific effect on gene expression, the fact that the superenhancer deletion so closely phenocopies the HOXA2 null mouse the likelihood that one enhancer might show a dramatically different phenotype is low.

3. Does deleting individual enhancers or even the whole 600Kb region change genome architecture near Hox genes? The authors should perform **Hi-C or 4C-seq in the deletion mutant and potentially in the individual enhancer mutants**.

We detailed this in the previous Figure 7 supplement 2 but have updated this to demonstrate more clearly. Specifically, in the revised version, Figure S22 contains a cartoon illustration to clarify the interactions of interest and the predicted shift in TADs created by the deletion.

Reviewer 3

Major comments

1. Figure 2B. Within the SE region, can the authors identify individual enhancers from the epigenomic profiling data? Are some of these active enhancers active by luciferase assay in human CNCCs?

In Figure 4 we have performed chromatin state and superenhancer calls for human CNCCs. These indicate similar patterns of chromatin activation that are more comparable to the primary tissue than the artificial luciferase assay.

2. The CNVs described in Figure 2 Supp1 are extremely large, impacting many genes including the HOXA gene cluster. Caution should be taken here, and the DECIPHER data doesn't support the statement "these findings strongly suggest that this superenhancer region is specific for craniofacial development and associated with craniofacial abnormalities" as the patient phenotypes are variable (with or without craniofacial abnormalities) and the CNVs impact many genes in addition to the SE region. This is discussed later in the manuscript, but is perhaps misleading in the earlier discussion of this data.

The analysis of the DECIPHER database has now been moved to the section describing the two patients with copy number variants (loss and gain). We also have tempered the discussion of this data and focus on the new copy number variants we have identified.

3. The logic for deleting the entire TAD in Figure 4 is somewhat unclear. By designing the guides either side of a predicted TAD boundary, the locus architecture may in fact be more greatly perturbed. More targeted deletion of the super enhancer (or elements within) would make interpretation of the results less ambiguous. Could the authors discuss further why they believe they are able to delete the enhancer cluster in mouse but not human? Are Hoxa mutations tolerated in hESCs?

While we cannot answer the question directly, our findings in the human fetus and human patient indicate dosage of *HOXA* gene expression driven by this region have much more deleterious outcomes in humans than in mice. Loss of one copy of the GCR in mice has no observable phenotype and recent work by Kessler et al suggest that phenotypes are only observed in a sensitized background.

4. What was the impact of heterozygous SE deletion in human CNCCs on expression of the HOXA genes? Are there SNPs within the HOXA genes that allow you to look at allele-specific expression from the wildtype versus SE-mutant alleles by qPCR or ddPCR?

We were able to use RNA-seq data to look at allele-specific expression in the hemizygous inversion line. These results are now presented in Figure 4 Supplement 7.

5. "All clones we obtained grew relatively normally in ESC culture conditions and we did not notice any major difference in morphology or gene expression". The gene expression analysis appears rudimentary in Figure 4 Supplement 3-4 – more sophisticated differential gene expression analysis and presentation, e.g. using DEseq and plotting base-mean expression versus foldchange with indication of significance would be more illustrative of the change of magnitude of expression (for example similar to the plot the authors have shown in Fig7A). Which classes of genes are changing their expression, if any?

We have extended our analysis of the hESC and CNCC cultures. Including GO terms enriched in the few differentially expressed genes and whether they are potentially related to *HOXA* signaling. These results are reported in Supplemental Table 10 and Figure S14.

6. "Given these results we reasoned that *HOXA* gene expression might be maintained at normal levels in these cell lines through upregulation of the cluster that remained in cis with the superenhancer region". This statement is ambiguous – are the authors suggesting that the SE activity increases with loss of one copy of the region? This could be tested by H3K27ac ChIP, and by allele-specific PCR strategies mentioned above. It would

be interesting to determine if inversion of the locus leads to an upregulation of the HOXA genes compared to the heterozygous lines given that you observe increased long-range interactions across the domain for the INV line.

A new supplemental figure showing boxplots of all the regional genes for the INV CNCC vs WT CNCC shows that expression of some HOXA genes looks greater in INV than WT, however no HOXA genes or genes in the region of the inverted superenhancer reached statistical significance.

7. “the robust expression of HOXA despite these changes precluded us from making any determination of the role of this region in craniofacial related biology”. The authors haven’t illustrated that HOXA gene expression isn’t significantly changed in the mutant CNCCs, a plot for this (RNA-seq or qPCR) would be informative.
See above, as the boxplots and statistical notation in Figure S11 address this concern.

9. The phenotypes of GCR deletion are not fully penetrant, does this correlate with a variable impact on Hoxa gene expression earlier in development between embryos? **Are the Hoxa2^{-/-} line phenotypes fully penetrant?**

This is a very good observation. We have made sure to clearly re-state where relevant the fact that Hoxa2^{-/-} phenotypes are not fully penetrant.

10. Figure 7A. What percentage of Hoxa2, Hoxa3 and Hoxaas2 expression is lost in the GCR mutant embryos?
No other gene expression changes are detected indicating that this effect is direct. **It may be interesting to look at a later stage of development to determine the impact on expression of Hoxa target genes.**
We agree this is an interesting concept but we focused our efforts on iPSC experiments from the GCR duplication patient.

11. Figure 7 Supplement 1. Several Hoxa genes appear to be impacted in the GCR homozygous deletion lines in limb tissue. Are there other limb enhancers that may be deleted in the large GCR deletion?
We have detailed all VISTA enhancers tested in all gene deserts in Supplemental Table 6. Elements hg1600, hg1465, and mm406 in addition to HACNS50 are reported to have both limb and craniofacial activity.

12. For Figure 7 Supplement 2. **Could the authors provide images of the HiC data rather than schematics.**
HiC data images are now incorporated into the updated version of this supplementary figure, now numbered as Figure 7 Supplement 5.

13. Figure 8 Supplement 2. **How do the 550kb deletion coordinates compare to the GCR deletion and to the SE you describe?**
That is illustrated in figure 8, the bar labeled de novo deletion. Maybe it could be brought next to the SE bar for better effect.

Minor comments

14. It would be informative to see the **intersection of the SEs detected across the 17 human craniofacial samples as an Upset plot or Venn diagram to know how stage-specific the elements are that have been detected.**

Given the large number of superenhancer regions identified in each craniofacial sample, we have compared superenhancer regions among the samples using Jaccard similarity. This is illustrated in Figure S1. We have found that, similar to gene expression data for craniofacial tissue, superenhancer similarities group into early and later stage clusters.

15. **An introduction to the ChromHMM states would aid the reader starting from Figure 2, including perhaps a key.**
We have added a key to figure 2.

16. Several of the enhancers in Figure 2C appear to be from mouse, **it would be useful to have an indication of the conservation of these elements to the human sequence.** Are all of the tested enhancers overlapping with predicted enhancer elements from your epigenomic profiling in human craniofacial tissues? **It may be useful to see the raw ChIP-seq traces for these enhancers.**
All mouse enhancer coordinates have been lifted over to human and thus are conserved. We have detailed chromatin state calls from both mouse and human in the orthologous windows in Figure S15. We have also

made comparisons between human craniofacial chromatin states and mouse face and brain in Figure S16.

17. It would be useful to have the SE region also marked on Figure 2 Supplement 1b along with the enhancer regions of interest within the SE. Which of these enhancers overlaps with the gnomAD variant-depleted region?

Could include SE depiction and validated enhancer locations.

18. The text mentions 10 patients with a single CNV at the HOXA locus which may be causative – could these be indicated in Figure 2 Supp1 – **how many of these are the patients with a craniofacial phenotype?**

We have clarified the language here and added more supplemental tables and figures:

“Of the 21 Individuals within the DECIPHER Database (deciphergenomics.org; Firth et al., 2009) with copy number variants (CNVs) overlapping the region (chr7:25,580,400-25,849,400;hg19). 14 of these had reported phenotypes, 10 of which had some type of craniofacial abnormality, a significantly higher incidence compared to the DECIPHER database as a whole (Figures S23 and S24, and Supplemental Table 13).”

19. **A comparison of the craniofacial tissues in which the tested enhancers from the HOXA SE are active (Figure 2C) to phenotypes detected in patients from DECIPHER could be informative.**

While this is an interesting point the enhancer assays are largely qualitative and the images available do not provide clear tissue or structure resolution. Furthermore the phenotype descriptions in DECIPHER are also not very precise preventing a robust analysis or interpretation.

20. Figure 2 Supp 3 is very hard to interpret as the boxplots are so thin. **Perhaps a separate plot per gene would be easier to read.**

We have altered this figure and provide a full size landscape version in the revised supplement (new Figure S5).

21. Figure 3. **It would be useful to indicate the regions that interact more strongly in CNCC versus ESC. Perhaps by subtraction maps.**

This aspect of the data now appears in revised Figure 4.

22. Figure 4B. “these closely spaced CTCF binding sites directly coincided with the boundary between strongly active and strongly repressed chromatin signals in both CNCCs and primary craniofacial tissues”. **CTCF sites don’t appear to be annotated in the figure.**

CTCF sites have been integrated into Figure 4 and a more detailed annotation is provided in figure S13.

23. Figure 7B shows rather variable expression for several genes and it is difficult to interpret. **Another representation of this data may be easier to interpret, e.g. boxplot or volcano plot with indication of significant changes? They are indeed variable and thus none of them are significantly different. We provide the heatmap to give some sense of changes we observed but lacked consistency.**

Reviewer comments, second version:

Reviewer #1 (Remarks to the Author: Overall significance):

1. In the original ‘super-enhancer’ papers (Whyte et al, 2012; Hnise et al, 2013), the median ‘super enhancer’ size is in the range of tens of kbp. ~600kb is close to the size a typical TAD and therefore appears too large. If it contains other functional elements, like structural elements, suppressor elements et al, deleting the entire 600kb would confound the proper interpretation. In fact, in the new Supplementary Figure 22, the authors found a shift in the TAD after deleting this large element, suggesting some structural changes that could impact gene regulation beyond enhancers. While I agree that the relatively large number (72 or 119) of putative regulatory elements within this region could make it tedious/time consuming to deconvolve the function of individual elements, the authors may still need to take some efforts to make smaller truncations--- such as two ~300kb, then ~4 150kb et al to further narrow down the bona fide enhancer region. This concern also echos that from Reviewer 2 and Reviewer 3. Such experiment could use CNCC

cells, even heterozygous deletion could still be informative. In fact, the massive parallel synthesis of oligos and reduction of cost has made the CRISPRi experiments more affordable and feasible to identify functional elements. If this experiment is not technically feasible for the investigator's lab, I am fine with some efforts to make smaller deletions and discussion in the text that more work is needed to identify individual enhancer contribution in future work.

2. Fine with the response.

3. Satisfied with the response.

4. While the reviewer agrees that the enhancer-promoter mode of contact is still a highly intriguing and debatable question in the field, I am asking what accounts for the 'higher contact frequency' in the context of CTCF site or domain inversion. 3D DNA FISH will provide an orthogonal way further understand the structural variations they created by CRISPR and observed by HiC, at single-cell resolution. Given their domain inversion is large, creating a ~600kb difference, 3D DNA FISH should have enough sensitivity to detect the change. This should provide additional mechanistic understanding on how putative change in loop formation (according to loop extrusion model) as a result of CTCF site inversion links to enhancer-gene contact. Even there is no change in terms of 3D distance measured by FISH, the authors could still cite those new studies (nonlinear relationship from Zuin et al).

Reviewer #2 (Remarks to the Author: Overall significance):

The authors have done a reasonable amount of work in revising their manuscript. I have one remaining question: Specific enhancers that affect HoxA genes have been published before. Even within the 600Kb region this manuscript focuses on, the authors identified at least 6 enhancers in Fig. 2. It is highly possible that targeting known enhancers has similar or even stronger effect compared with deleting or inverting the 600Kb selected region. The authors did not make effort to generate deletions suggested in the original review. At the minimum they should discuss previous enhancer deletion studies and clarify the difference between this study and previous results.

Reviewer #3 (Remarks to the Author: Overall significance):

The authors have included new analysis and updated figures in response to many of my queries. However, the rebuttal in general is challenging to follow as the authors have not included the new text in the rebuttal nor highlighted the textual changes in the updated manuscript. Also, the rebuttal does not appear to be carefully proof-read, as the naming of the supplementary figures are inconsistent with the rebuttal text, e.g. Figure 4 Supplement 7 from the rebuttal appears to be Figure S9.

Additional comments about the authors' response:

Point 5. The authors refer to GO term analysis in Figure S14 which doesn't appear to be present in the figure.

Point 6. The authors refer to a figure without referring to the figure number. Is this S11?

Point 12. Again, the supplementary figure numbering appears incorrect.

Point 13. Did the authors make a change to the figure here? The rebuttal response appears to be a suggestion, not a definitive response.

Point 17. This comment appears to be a note and not a formal response to the comment.

Author rebuttal, second version:

We thank the reviewers for re-examining our work. We are pleased that they recognize the additional work that was added. Recent work from Kessler et al addresses some of the issues raised by all three reviewers related to additional deletions. Thus we feel that such additional experiments do not substantially add to findings of this paper. Instead we are more confident that ever that this region is important for normal mouse development. We will point out that we added additional human patient data that bolsters the idea that this region is also important for normal human development. This extends the findings of Kessler et al and warrant publication to demonstrate this particular importance in human. We address specific comments laid out by each reviewer below.

Review 1 comments:

In the original ‘super-enhancer’ papers (Whyte et al, 2012; Hnisz et al, 2013), the median ‘super enhancer’ size is in the range of tens of kbp. ~600kb is close to the size a typical TAD and therefore appears too large. If it contains other functional elements, like structural elements, suppressor elements et al, deleting the entire 600kb would confound the proper interpretation. In fact, in the new Supplementary Figure 22, the authors found a shift in the TAD after deleting this large element, suggesting some structural changes that could impact gene regulation beyond enhancers. While I agree that the relatively large number (72 or 119) of putative regulatory elements within this region could make it tedious/time consuming to deconvolve the function of individual elements, the authors may still need to take some efforts to make smaller truncations--such as two ~300kb, then ~4 150kb et al to further narrow down the bona fide enhancer region. This concern also echos that from Reviewer 2 and Reviewer 3. Such experiment could use CNCC cells, even heterozygous deletion could still be informative. In fact, the massive parallel synthesis of oligos and reduction of cost has made the CRISPRi experiments more affordable and feasible to identify functional elements. If this experiment is not technically feasible for the investigator’s lab, I am fine with some efforts to make smaller deletions and discussion in the text that more work is needed to identify individual enhancer contribution in future work.

We will point out that these two papers use different data types for calling of superenhancers. Whyte et al originally used MED1 ChIP-Seq signal while Hnisz et al and most subsequent papers use H3K27ac. The Whyte paper identified a relatively small number (<300 genome-wide) and were smaller in size than those in subsequent papers. We have added an additional panel to Figure S2 to demonstrate that our superenhancer size distributions are not significantly different than dozens of other tissues that use H3K27ac data. Furthermore we state specifically on page 6 of the manuscript:

“Due to the high proportion of enhancer segments with confirmed craniofacial activity, we chose to focus on the gene desert located on chromosome 7. This chromosomal segment contained three regions identified by ROSE as superenhancers active in human embryonic craniofacial tissue (**Figure 2a**). The superenhancer regions between chr7:25,580,400-25,880,000 (hg19) are unique to human embryonic craniofacial tissue, not having been identified as such in human embryonic heart tissue (VanOudenhove et al., 2020) or any of the 102 human tissues and cell lines analyzed by dbSuper (Khan and Zhang, 2016 and <https://asntech.org/dbsuper/index.php>) (**Figure 2b**).”

Thus while we agree that an individual superenhancer call being 600kb would be quite large we focused on this region because it contained three superenhancer regions. Our deletion and those recently described by Kessler et al directly address the issue raised by Reviewer 1. We show these deletions in several supplemental figures and add discussion of how these help to identify a potential minimal critical region. There remain dozens of potential enhancer modules in the HIRE1 region alone that would require much more time and effort to explore. In the revision we tried to reframe our findings to focus on the larger region and its impact on normal human development. However, the reviewer did not acknowledge our inclusion of new human patient data that strongly supports that copy number variation in this region deleterious in humans. While CRISPRi based experiments in CNCCs could explore some aspects of gene regulation, they cannot be further

developed into a face or skull where the major phenotypes related to this region are observed.

While the reviewer agrees that the enhancer-promoter mode of contact is still a highly intriguing and debatable question in the field, I am asking what accounts for the ‘higher contact frequency’ in the context of CTCF site or domain inversion. 3D DNA FISH will provide an orthogonal way further understand the structural variations they created by CRISPR and observed by HiC, at single-cell resolution. Given their domain inversion is large, creating a ~600kb difference, 3D DNA FISH should have enough sensitivity to detect the change. This should provide additional mechanistic understanding on how putative change in loop formation (according to loop extrusion model) as a result of CTCF site inversion links to enhancer-gene contact. Even there is no change in terms of 3D distance measured by FISH, the authors could still cite those new studies (nonlinear relationship from Zuin et al).

As mentioned above we included additional human patient data that further supports our assertion that proper dosage of this region is essential for normal human development. While we agree this is an interesting concept that this region could be used to study, this is beyond the scope of this work. We did add discussion of the concept of linear versus nonlinear relationships of enhancer activities.

I am not sure “and is true generally from other studies, super-enhancers typically contain one or more genes” is accurate. In many early studies (Hnise et al, 2013; Whyte et al, 2012; Li et al, 2014; Zhou et al, 2014; Xie et al, 2017) super enhancers do not contain genes. Even in this manuscript, I do not see a gene that overlaps with the enhancer they chooses to work with (Fig.2a-b). Moreover, Reviewer2 has the same concern as me regarding this study failing to identify the constitutive enhancer elements. In alignment with Reviewer2, the ~600kb region they choose to work is too large. In the original ‘super-enhancer’ papers (Whyte et al, 2012; Hnise et al, 2013), the median ‘super enhancer’ size is in the range of tens of kbp. ~600kb is close to the size a typical TAD. While I agree that the large number (72 or 119) of putative elements within this region could make it technically challenging (if not impossible, e.g., CRISPRi) to pinpoint the contribution of individual element, the authors may still need to take some effort to make smaller truncations---such as two ~300kb, then ~4 150kb et al to further narrow down the bona fide enhancer region. This will actually make their argument stronger about enhancer regulation of tissue-specific genes (HOXA) and more mechanistic insights on enhancer regulation.

The reviewer is simply incorrect in their assertion of overlapping of genes. While the figures in those papers highlight regions that do not overlap at TSS, examination of their data as well as superenhancer calls from dozens of other tissues show the majority do. We detailed this originally in Figure S2. The size assertion made here is also incorrect. We have added a panel to Figure S2A that clearly shows that superenhancer calls can extend to hundreds of kilobases. Furthermore, the region we identified harbors multiple superenhancers that we hypothesized could function as a single unit. Our results combined with Kessler et al refine this region to an approximately 175 kb in mouse. We discuss these new findings and show the deletions in several supplemental figures. However our human patient data suggests different regions, particularly the TAD boundary, could also be important in human.

Reviewer #2 (Remarks to the Author: Overall significance):

The authors have done a reasonable amount of work in revising their manuscript. I have one remaining question: Specific enhancers that affect HoxA genes have been published before. Even within the 600Kb region this manuscript focuses on, the authors identified at least 6 enhancers in Fig. 2. It is highly possible that targeting known enhancers has similar or even stronger effect compared with deleting or inverting the 600Kb selected region. The authors did not make effort to generate deletions suggested in the original review. At the minimum they should discuss previous enhancer deletion studies and clarify the difference between this study and previous results.

We have added more discussion of the results of Kessler et al and show these deletions on several supplemental figures for comparisons. We also discuss the concept of linear versus synergistic activities of multiple enhancers with respect to superenhancer function.

Reviewer #3 (Remarks to the Author: Overall significance):

The authors have included new analysis and updated figures in response to many of my queries. However, the rebuttal in general is challenging to follow as the authors have not included the new text in the rebuttal nor highlighted the textual changes in the updated manuscript. Also, the rebuttal does not appear to be carefully proof-read, as the naming of the supplementary figures are inconsistent with the rebuttal text, e.g. Figure 4 Supplement 7 from the rebuttal appears to be Figure S9.

We apologize for not streamlining the supplemental figure calls from the main manuscript into the rebuttal. We have checked all points below and the figures and tables are correct in the revised manuscript that reviewers received.

Additional comments about the authors' response:

Point 5. The authors refer to GO term analysis in Figure S14 which doesn't appear to be present in the figure.

The GO terms are in table S10 and the correct figure call is S9. These were correctly referenced in the main manuscript Page 9 lines 343 to 366 but were unfortunately not updated in the reviewer response document.

Point 6. The authors refer to a figure without referring to the figure number. Is this S11?

Yes this is figure S11.

Point 12. Again, the supplementary figure numbering appears incorrect.

Again we apologize for this oversight. The correct figure label is S22.

Point 13. Did the authors make a change to the figure here? The rebuttal response appears to be a suggestion, not a definitive response.

We have added an additional panel to Figure S26 that indicates the de novo deletion, the orthologous deletion described in this work, and the two separate deletions described by Kessler et al.

Point 17. This comment appears to be a note and not a formal response to the comment.

We apologize for not providing the updated figure. We have added the craniofacial superenhancer regions, individual enhancer calls, and regions deleted in this study and Kessler et al to Figure S23B.